



# Updated algorithmic climate change functions (aCCF) V1.0A: Evaluation with the climate-response model AirClim V2.0

Sigrun Matthes[1*], Simone Dietmüller[1], Katrin Dahlmann[1], Christine Frömming[1], Patrick Peter[1], Hiroshi Yamashita[1], Volker Grewe[1,2], Feijia Yin[2], Federica Castino[2]

[1]DLR, Institute of Atmospheric Physics, German Aerospace Center, Oberpfaffenhofen, Germany
[2]TU Delft, Faculty of Aerospace Engineering, Delft University of Technology, Delft, Netherlands

*Correspondence to*: Sigrun Matthes (Sigrun.matthes@dlr.de)

**Abstract.** Aviation aims to reduce its climate effect by exploiting the potential of identifying alternative climate optimized aircraft trajectories. Such climate-optimized trajectories require a dedicated meteorological service in order to inform on those regions of the atmosphere where aviation emissions have a large effect on climate, for example, by contrail formation or nitrogen-oxide ($NO_x$)-induced ozone formation. With the algorithmic Climate Change Functions (aCCFs) prototypes of a mathematical formulation for the temporal and spatial climate effects of aviation emissions in the atmosphere is provided, which relies solely on numerical weather prediction at the time and location of emissions. Based on the recently published consistent set of aCCF-V1.0, we here introduce newly derived calibration factors for the individual non-$CO_2$ effects of aviation ($NO_x$, water vapour, contrail cirrus) and establish version V1.0A of aCCFs (aCCF-V1.0A). ACCF-V1.0A represents an updated formulation of aCCF while exploring the current level of scientific understanding of individual climate effects of aviation emissions by evaluating quantitative estimates of climate effects with the state-of-the-art climate-response model AirClim V2.0. Individual scaling factors (i.e. AirClim calibration factors) are provided for the respective non-$CO_2$ effects comprising contrail cirrus, water vapour and $NO_x$-induced climate effects on ozone and methane, resulting uniformly in lower estimates in aCCF-V1.0A for all species compared to the earlier version aCCF-V1.0.

## 1. Introduction

Global aviation significantly contributes to anthropogenic climate change through $CO_2$ and non-$CO_2$ emissions. Non-$CO_2$ emissions comprise mainly water vapour ($H_2O$), nitrogen oxides ($NO_x$), sulphur oxides and soot. Not all non-$CO_2$ emissions have a direct effect on climate. For example. $NO_x$ emissions are not radiatively active themselves, but they are responsible for the chemical production of the greenhouse gas (GHG) ozone ($O_3$) and the destruction of the GHG methane ($CH_4$). Moreover, by the $NO_x$-induced methane decrease long-term background ozone is reduced (primary mode ozone, PMO, see e.g. Stevenson et al., 2004) as well as stratospheric water vapour (Myhre et al., 2007). Furthermore, induced by water vapour and particle emissions, contrails and contrail-cirrus can form and alter the radiation budget.

In contrast to aviation's $CO_2$ emissions, the non-$CO_2$ effects show a strong dependence on the local meteorological and chemical background conditions. Accordingly, non-$CO_2$ effects depend on the geographical location (longitude, latitude, altitude), time of the aircraft emission (e.g. Gauss et al., 2005; Stuber et al., 2006; Köhler et al., 2008, Frömming et al., 2012)



and the prevailing weather situation. This relationship can be exploited by identifying alternative flight trajectories which avoid sensitive regions where the climate effects are high (e.g. Matthes et al, 2021). Earlier studies regarding climate impact mitigation through operational means investigated permanent changes in flight altitudes or lateral changes of routes (e.g., Sausen et al., 1998, Fichter et al., 2005, Rädel and Shine, 2008), or investigated the avoidance of contrails and contrail-cirrus through small changes in flight levels if flying through contrail forming areas (e.g., Mannstein et al., 2005, Chen et al., 2012, Teoh et al., 2020) or horizontal trajectory alterations (Sridhar et al., 2011) or both (e.g., Zou et al., 2016, Hartjes et al., 2016; Yin et al., 2018). The spatial and temporal variability of aviation $NO_x$ emissions has been studied by permanent alterations of flight altitudes or routes (e.g., Gauss et al., 2006, Köhler et al., 2008), while the altitudinal and latitudinal variability of annual mean aviation effects was studied systematically, e.g. by Grewe and Stenke (2008). However, research on weather-dependent climate-optimized aircraft trajectories requires detailed information on aviation non-$CO_2$ effects provided during the flight planning process.

Various concepts exist in order to calculate climate effect related to such spatially and temporally varying non-$CO_2$ effects, and we distinguish three different types here. One type of study relies on explicit numerical simulations applying alternative emission scenarios in a comprehensive chemistry-climate model, e.g. solely for $NO_x$-induced effects in a multi-model study (Sovde et al. 2014), or for non-$CO_2$ effects of contrails, nitrogen oxides and water vapour (Frömming et al., 2012), or additionally comprising aerosol-induced indirect effects (Matthes et al., 2021). The second type of study relies on temporally averaged response surfaces which have been developed in general with the help of more comprehensive climate system modelling. Such response surfaces can be seen as short-cut between aviation emissions and associated climate effects. Depending on the individual study concept, a certain number of parameters are considered in order to represent dependencies: typically, altitude, geographic location, and in some cases time of emission. In an earlier work, a climate response model for calculating climate effects of $NO_x$ and aircraft induced cloud effects solely as a function of altitude of emission has been presented (Schwartz-Dallara et al., 2011). A more comprehensive climate-response model (AirClim V2.0) allows to estimate climate effects while additionally considering the latitudinal dependency (Dahlmann et al., 2016). AirClim relies on the perturbation approach for $NO_x$-inducing effects on ozone and methane, and water vapour direct effects. For contrail cirrus effects the longitudinal dependency is furthermore considered. In the third type of study, an additional time dimension is introduced to consider the influence of the synoptic situation. In order to provide 4-dimensional information on these spatially and temporally varying non-$CO_2$ effects, climate change functions (CCFs) were developed, which derive such information from detailed simulations with a comprehensive atmospheric chemistry climate model system (Matthes et al., 2012; Grewe et al., 2014a; Frömming et al., 2021). These CCFs introduce aviation-induced climate effects per emission for $NO_x$, contrails and water vapour effects for the North Atlantic Flight Corridor (NAFC) for eight specific days that consider representative weather types in summer and winter (Frömming et al., 2021), relying on the tagging contribution approach. Grewe et al. (2014b) used CCFs for air traffic re-routing for one winter case study day. However, CCF calculations within a chemistry climate model are impractical from a computational perspective and thus they cannot be used operationally for trajectory optimization. For this reason, an efficient implementation concept for trajectory planning tools, the algorithmic Climate Change Functions (aCCFs),





was defined and initially introduced (Matthes et al., 2017; van Manen and Grewe, 2019). The overall concept of aCCFs development relies on statistical methods which link and correlate non-$CO_2$ climate effects as quantified by the CCFs (explicitly calculated in comprehensive numerical chemistry climate model simulations) to local meteorological conditions. Therefore, the strength of the aCCF is that their implementation solely requires the local state of the atmosphere as input data.

Van Manen and Grewe (2019) have published the first formulation of aCCFs addressing the $NO_x$-induced $O_3$ and $CH_4$ effects and the climate effects from water vapor. Yin et al. (2022) presented the first consistent set of aCCFs for $NO_x$-$O_3$, $NO_x$-$CH_4$, $H_2O$, $CO_2$, and contrail cirrus effects. The technical implementation of aCCF-V1.0 is provided in the open source Python library CLIMaCCF V1.0 (Dietmüller et al., 2022), and in the submodule ACCF 1.0 (Yin et al., 2022) of the global chemistry climate model EMAC (ECHAM5 version 5.3.02, MESSy version 2.54.0, Jöckel et al., 2010) in the T42L31ECMWF-

resolution. These aCCFs are response functions specifically designed to enable the implementation of climate-optimized flight trajectories. For an initial proof of concept, Yin et al. (2018) and Rao et al. (2022) applied an atmospheric chemistry model chain within the EMAC framework to evaluate the effectiveness of $NO_x$-$O_3$ aCCFs using their climate effect information for trajectories optimizations.

Due to the respective level of scientific understanding of aviation induced climate effects, uncertainties exist in their
quantitative estimates (see e.g. confidence intervals of radiative forcing estimates in Lee et al., 2021 and Grewe et. al., 2017). Various approaches on how to deal with these uncertainties have been performed (e.g. Dahlmann et al., 2016; Grewe et al., 2021; Brazzola et al., 2022). A formulation of the aCCF algorithms, that explores the range of uncertainty of aCCF estimates is yet missing and thus the room for a calibration process is open. Specifically, in this paper we evaluate results from two distinct response modelling concepts, where one concept resolves variation due to the synoptic situation (aCCF) and another
concept relies on averaged annual mean response surfaces (AirClim), to perform a calibration of the algorithms.

The objective of this paper is to document recent evaluation processes performed within the EU project FlyATM4E resulting in a new version of aCCFs: aCCF-V1.0A. This new version of aCCFs is calibrated to the results of the climate-response model AirClim. The aCCF-V1.0A can be seen as one realization within the range of plausible values (event horizon) considering the current level of scientific understanding of climate effects of aviation emissions and their associated uncertainty. This was
done by evaluating and comparing the climate effects of aviation, resulting from aCCF-V1.0, with an alternative numerical model; here we used the state-of-the-art climate-response model AirClim. A detailed description, visualization, and application of aCCF-V1.0A are provided.

The paper is structured as follows: Section 2 provides the workflow applied to develop calibrated to AirClim (aCCF-V1.0A) and the underlying methods. Section 3 presents individual evaluations performed and introduces aCCF-V1.0A, while
describing the update from aCCF-V1.0 to aCCF-V1.0A in detail and the final choice of calibration parameters. In Section 4 we apply the aCCF-V1.0A to a typical weather situation in summer and winter. Moreover, we provide a comparison between aCCF -V1.0A and aCCF-V1.0. A discussion, describing the major strength of aCCF-V1.0A follows in Section 5, before the conclusion and outlook are given in Section 6.





## 2. Methods

The calibration concept relies on combining estimates of non-$CO_2$ effects from aCCF-V1.0 with estimates from the climate-response model AirClim. We use V1.0 of the algorithmic climate change functions as described in Yin et al. (2022) which represent a prototype for a quantitative estimate of the climate effect of aviation emissions released under specific atmospheric conditions. The formulation of aCCFs-V1.0 quantifies the climate effect of an emission in terms of the average temperature response (ATR), while conversion factors to other physical climate metrics are available. The physical climate metric ATR

measures temperature effects over a chosen time horizon, e.g. 20, 50 or 100 years.

The mathematical formulation given in aCCF-V1.0 (Yin et al., 2022) provides a numerical value which represents the statistical mean value of the probability distribution per individual $CO_2$ and non-$CO_2$ climate effects; hence it relates to a confidence interval of possible values. From a statistical point of view, each value in this interval is possible. Instead of choosing the mean value to calculate the climate effect, it is equivalent to selecting another value from this interval. Consequently, for our

calibration process, we explore in this study the underlying range of uncertainties, and align the quantitative estimate of aCCFs to alternative modelling studies with the climate-response model AirClim.

### 2.1 Workflow of the calibration experiment

In order to generate this updated version of aCCF-V1.0A, we developed a workflow which relies on comparing climate effects from a traffic emission inventory calculated with (1) aCCFs, and (2) the climate-response model AirClim (Fig. 1). With this

evaluation, we update the formulas provided in Yin et al. (2022) (aCCF-V1.0) to aligned the strength of individual non-$CO_2$ effects. Fig. 1 illustrates the workflow which allows to provide the AirClim calibrated aCCFs (aCCF-V1.0A). The AirClim calibrated aCCFs calculate the effects in such a way that the estimated climate effect is corresponding to the AirClim estimate.



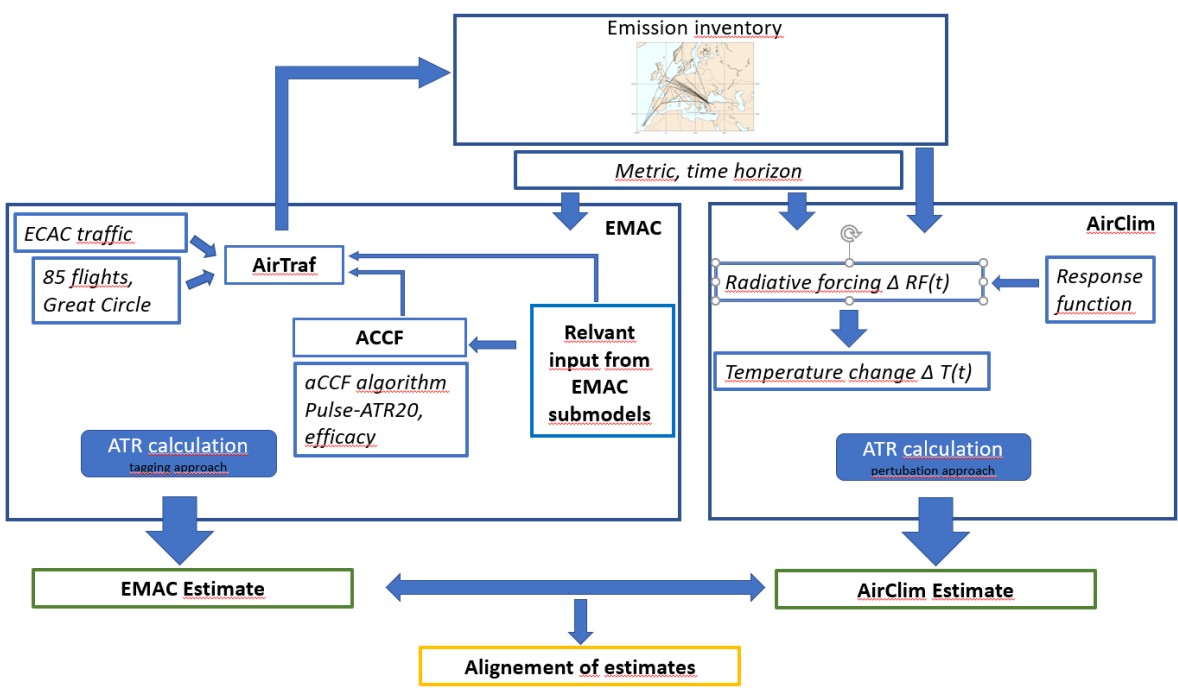

**Figure 1: Schematic workflow describing the calibration approach leading to aCCF-V1.0A.**

In brief, our workflow starts with a representative sample of European air traffic (city pairs) from which EMAC/AirTraf constructs a set of aircraft trajectories, resulting in a European emission inventory. For the construction of aCCF-V1.0A, the climate effects of this European air traffic sample are evaluated by the prototype aCCFs-V1.0. This numerical calculation is performed online within the modular chemistry-climate model EMAC, by coupling the air traffic simulator AirTraf (version 2.0, Yamashita et al., 2016, 2020, see Fig. 1) to the submodel ACCF (Yin et al., 2022). Contrail distance is calculated by

multiplying the flown distance with the potential contrail cover as quantified by the submodel CONTRAIL (version 1.0, Frömming et al., 2014). During a one-year simulation, climate effects are calculated for each time step, and finally summed up in order to calculate the total annual climate effect, comprising $CO_2$ and non-$CO_2$ effects. Additionally, EMAC/AirTraf constructs an annual emission inventory for this traffic sample, which is subsequently used as an input to AirClim (Dahlmann et al., 2016). Consistently, the climate-response model AirClim calculates the climate effect for the same European air traffic

sample assuming constant emissions on an annual basis. By comparing both estimates on an annual basis, a set of scaling factors (i.e., AirClim calibration factors) is identified, which are then applied to aCCF-V1.0, to construct aCCF-V1.0A.

## 2.2 Trajectory planning in the global chemistry-climate model EMAC/AirTraf/ACCF – using aCCF 1.0

A submodel enabling aircraft trajectory planning is implemented in the chemistry-climate model EMAC (Jöckel et al., 2010) with the coupling to the submodel AirTraf (Yamashita et al., 2016, 2020; see Fig. 1). EMAC is a numerical chemistry and

climate simulation system that includes submodels describing tropospheric and middle atmosphere processes and their




interaction with oceans, land, and influences coming from anthropogenic emissions. It comprises the second version of the Modular Earth Submodel System (MESSy2) to link multi-institutional computer codes, in which the core atmospheric model is the fifth generation European Center Hamburg general circulation model (ECHAM5, Roeckner et al., 2006). We use a horizontal resolution of T42, with 31 hybrid vertical pressure levels up to 10 hPa (~30 km, T42L31ECMWF) and a time step

of 20 minutes, with meteorology nudged to ERA5 reanalysis data as boundary conditions. AirTraf allows to calculate aircraft trajectories for given city-pairs with respect to dedicated routing strategies, considering meteorological conditions calculated by ECHAM5 online. Associated with these identified aircraft trajectories, the climate effect from aviation emissions along these trajectories is calculated from the submodel ACCF version 1.0 of EMAC (Yin et al., 2022). ACCF employs the aCCFs, that provide spatially and temporally resolved information on the climate effects of aviation emissions to quantify $CO_2$ and

non-$CO_2$ effects (Section 1). Specifically, they allow identifying regions of the atmosphere where aviation emissions induce a strong climate effect, e.g. via the formation of warming contrails or the production of radiatively active species like ozone. The recent publication of Yin et al. (2022), provides the mathematical formulation of a first consistent set of individual aCCFs (aCCF-V1.0), including $CO_2$ (a constant value in K/kg (fuel)), water vapour (K/kg(fuel)), $NO_x$ -induced ozone and methane (including primary mode ozone) (K/kg($NO_2$)) and contrail cirrus aCCFs (K/km). Thus, using these aCCFs, the estimated

climate effect of aviation emissions and their spatial and temporal variability is available for the model domain: Subsequently, this is provided to AirTraf in order to enable not only estimating climate effects for calculated aircraft trajectories, but also planning of climate-optimized flight trajectories. The combination of EMAC/AirTraf/ACCF is applied here to simulate flight trajectories as great circle routes between city pairs (equal to AirClim), considering the variability of synoptic weather patterns in a continuous representation of the global atmosphere, and to quantitatively assess total climate effect and overall

performance of flights.

**2.3 The non-linear climate-chemistry response model AirClim**

AirClim is a non-linear climate-chemistry response model to simulate the climate effect resulting from aircraft emissions (V2.0, Dahlmann et al., 2016). AirClim combines air traffic emissions with pre-calculated atmospheric impact data (Fig. 1).

For the pre-calculated data, climate–chemistry simulations are performed, with normalised emissions in idealised emission regions. For receiving the response surfaces for $NO_x$ emissions on the $O_3$ and $CH_4$ concentration, and for the impact of $H_2O$ emissions, the complex chemistry-climate model E39C/A (Stenke et al., 2008) was used. For the response surfaces of contrail induced cloudiness simulations with the ECHAM4-CCMod model (Burkhardt and Kärcher, 2011) were analysed. The climate effect of $CO_2$ is independent of the emission location due to its long lifetime. Therefore, a green function based on Hasselmann

et al. (1997) is used for the climate impact of $CO_2$. AirClim calculates the temporal development of concentration change, radiative forcing (RF) and near surface temperature change of the individual effects accounting for their different lifetime and dependency on the emission location.



## 2.4 Calibration experiment setup for non-CO₂ climate effects

In this section, we describe our calibration experiment, consisting of the comparison of the climate effect estimates obtained
using EMAC/AirTraf/ACCF and AirClim. To conduct this experiment, we consider 85 representative city-pair connections in
the European airspace, having departure times distributed throughout the day. This is a subset of an air traffic sample developed
during the ATM4E project (Matthes et al., 2020), which was obtained from an analysis of scheduled air traffic in the ECAC
(European Civil Aviation Conference) area selecting only the flights using A330 type of aircraft models. This sample of city-
pair connections consists of the coordinates of the departure and arrival airport, as well as the departure time.

In the first step of our experiment (Fig. 1), this sample of city-pairs is provided as input to AirTraf V2.0, and a one-year
simulation is carried out for the period from 1 December 2015 to 1 December 2016. To reduce the computational cost and the
degrees of freedom of the problem, great circles connecting origin and destination with 101 waypoints are used for each flight
trajectory; only the cruise phase is considered, with a ceiling height at flight altitude FL350 (10668 m, 240 hPa). This
simulation provides two main results: (1) the annual emission inventory for the simulated flights as input to AirClim V2.0, and
(2) the estimates of the climate effects of those flights, which are computed via coupling with the EMAC model and, in
particular, with the ACCF submodel applying aCCF-V1.0 (see Figure 1).

In the second step, the climate effects of this aviation emission inventory are calculated in a simulation with the climate-
response model AirClim. While AirClim is calculating climate effects on an annual basis, estimating the climatological mean
effect, aCCFs estimate the climate effect with a synoptic dependency. In order to have comparable results, we calculate the
impact with aCCF over a whole year. Both calculations apply the physical climate metric average temperature response over
a time horizon of 20 years, assuming pulse emissions (P-ATR20).

As a result of these two numerical simulations, we obtain two sets of P-ATR20 values for the individual non-CO₂ effects,
measuring the climate impact of contrails, NO$_x$ emissions via ozone or methane perturbations, and water vapour associated
with one year of a European air traffic sample. This comparison constitutes the basis for the calibration of aCCF formulas
towards the AirClim results. AirClim calibration factors are derived by dividing the estimates of individual non-CO₂ effects
calculated by EMAC and AirClim. Specifically, aCCFs are calibrated in such a way to the AirClim values, that the individual
annual total non-CO₂ climate effects are in line with the relative importance of individual non-CO₂ effects.

## 3. Mathematical formulation of aCCF-V1.0A

This section provides the quantitative estimates of the non-CO₂ effects as calculated by EMAC/AirTraf/ACCF and AirClim.
Table 1 provides the quantitative values of the individual climate effects (given as P-ATR20 in [K]) from NO$_x$-induced ozone
and methane (including PMO), water vapour and contrail cirrus, resulting from an emission inventory representing 85 city-
pair connections, calculated both with the EMAC/AirTraf/ACCF model (Section 2.2) and the state-of-the-art climate-response
model AirClim (see Section 2.3). A detailed description of the two underlying model setups is given in Section 2.4.





**Table 1: Individual aviation climate effects in calibration workflow: $CO_2$, $NO_x$-induced $O_3$, $NO_x$-induced $CH_4$(including PMO), $H_2O$ and contrail-cirrus effects. Individual climate effects are provided as P-ATR20 in [K] (without forcing efficacy) for annual totals of the year 2016 calculated by AirClim and EMAC/AirTraf/ACCF. The underlying traffic sample comprises 85 city-pair connections using great circle trajectories. Additional fuel use and emissions of $NO_x$, $H_2O$, flown kilometres, and contrail kilometres are provided.**

| P-ATR20[K] | $CO_2$ | $O_3$ | $CH_4$+PMO | $H_2O$ | Contrail-cirrus |
|---|---|---|---|---|---|
| AirClim | $2.16\ 10^{-07}$ | $9.68\ 10^{-07}$ | $-1.02\ 10^{-07}$ | $7.94\ 10^{-08}$ | $9.84\ 10^{-07}$ |
| EMAC/AirTraf/ ACCF | | $5.41\ 10^{-06}$ | $-1.74\ 10^{-06}$ | $1.31\ 10^{-07}$ | $1.51\ 10^{-06}$ |
| | **Fuel used [kg]** | **$NO_x$ [g $NO_2$]** | | **$H_2O$ [g]** | **Distance [km]** |
| Fuel/emissions | $2.894\ 10^8$ | $3.134\ 10^9$ | | $3.559\ 10^{11}$ | |
| Flown distance | | | | | $4.89\ 10^7$ |
| Contrail distance | | | | | $1.301\ 10^7$ (AirTraf) |

Comparing the quantitative climate effect estimates between AirClim and EMAC/AirTraf/ACCF reveals lower P-ATR20 values in the AirClim calculations for all individual species. Based on this comparison (Table 1), we derive AirClim calibration factors ($f_{AirClim}(i)$) for each individual climate effect i (with i={$O_3$,$CH_4$, $H_2O$, contrail-cirrus}). These calibration factors were derived by dividing the specific AirClim estimate by the EMAC/AirTraf/ACCF estimate ( $f_{AirClim}(i) =$ P-ATR20$_{AirClim}(i)$/P-ATR20$_{ACCF}(i)$) and are summarized in Table 2. Note that we apply identical calibration factors to the

effect of PMO as to methane, as this indirect chemical PMO effect is related to identical chemical mechanisms and their temporal evolution.

**Table 2: ACCF calibration factors, $f_{AirClim}$ (i), derived by quantitative comparision of individual climate effects calculated by AirClim and EMAC/AirTraf/ACCF (see Table 1).**

| | $O_3$ | $CH_4$+PMO | $H_2O$ | Contrail-Cirrus |
|---|---|---|---|---|
| $f_{AirClim}(i)$ | 0.179 | 0.058 | 0.641 | 0.333 |

We apply these AirClim calibration factors to the individual aCCF-V1.0(i), as we aim to make sure that the quantitative estimates have about the same magnitude as those from a climate-response model calculation. The formulation of the individual aCCF-V1.0A(i) is then given by simply multiplying the individual components of aCCF-V1.0(i) by the AirClim calibration

factors:

$$aCCF\text{-}V1.0A(i) = f_{AirClim}(i)*aCCF\text{-}V1.0(i) ,\qquad\qquad(1)$$

The value for $CO_2$ aCCF is identical to that of aCCF-V1.0 (Yin et al. 2022) with $7.48 \cdot 10^{-16}$ K/kg(fuel). The complete formulas are provided in Annex A.

## 4. Application of ACCF V1.0A on an individual summer day in Europe

To give an impression of the typical geographical aCCF-V1.0A distribution at a pressure level of 250 hPa over European air space, Fig. 2 shows individual patterns of water vapour aCCF, $NO_x$-induced aCCF (including ozone, methane and primary





mode ozone effect) and contrail-cirrus aCCF during the course of one example day, which is a typical summer day in the year 2018. ACCF patterns are calculated and shown every 6 hours between 14 June (12 UTC) and 15 June (12 UTC), using meteorological input from the ERA5 high resolution realization data set (Hersbach et. al. 2020), with a horizontal resolution

of 0.25°x0.25°. The weather situation is characterized by a strong zonal jet stream and a positive East Atlantic Oscillation index (classified as weather pattern S1 after Irvine et al., 2013). Note, that aCCFs are calculated here in terms of ATR20, employing a future business-as-usual emission scenario (F-ATR20) including efficacies of all species (for respective metric conversion factor and efficacy values see Yin et al., 2022; Dietmüller et al., 2022). Moreover, all aCCFs were converted to the same unit of K/kg(fuel) by multiplying the individual aCCFs by typical transatlantic fleet mean values of 13 g($NO_2$)/Kg(fuel)

(Graver and Rutherford, 2018) for $NO_x$ emission indices (in case of ozone and methane aCCF) and of 0.16 km/kg(fuel) (Penner et al., 1999) for flown distance per kg burnt fuel (in case of contrail aCCF). Next to the aCCFs, which represent individual effects, also merged aCCFs are shown (Fig. 2), which combine the non-$CO_2$ effects in one single aCCF (Dietmüller et al. 2022).

The water vapor aCCFs (Fig. 2a) show positive (warming) values, with only small variations during the course of one day

(time steps are given by six hours). ACCFs of water vapour are comparably small for the present situation and pressure level. The total $NO_x$-induced aCCFs (Fig. 2b) combine the positive (warming) values of the ozone aCCF and the negative (net-cooling) values of the methane and primary mode ozone (PMO) aCCFs. Overall, the total $NO_x$-induced aCCFs are positive and a zonal gradient can be observed with high positive values at lower latitudes (Fig. 2b). This can be explained by the higher ozone formation in lower latitudes (Rosanka et al., 2020). The temporal evolution of $NO_x$-induced aCCFs during the 24 hours

shown reveals only slight changes in structure and magnitude. Fig. 2c provides the contrail aCCFs during the course of the selected summer day. During day-time conditions, contrails can have both longwave and shortwave radiative effects depending on the zenith angle, thus the day-time contrails may have negative and positive values. In contrast, night-time contrail aCCFs always cause a warming effect with positive values. The negative shortwave effect is eliminated at night due to the lack of solar radiation, thus only the longwave effect remains. As contrail formation and contrail climate effects are very sensitive to

the background atmospheric conditions, contrail aCCFs show, as expected, a very large variability during the course of the 24 hours shown here. In addition to the aCCFs for individual effects, merged non-$CO_2$ aCCFs, which combine non-$CO_2$ effects are shown in Fig. 2d. Evaluating the individual aCCFs together with the merged aCCFs, it becomes clear that contrail cirrus aCCFs dominate the merged aCCF structure and magnitude in those regions where contrail cirrus effects appear. The water vapour aCCF can be considered negligible due to its low values.

In order to give an impression of the spatially and temporally resolved climate effects of aviation emissions in winter we also provide the temporal evolution of the aCCF patterns within 24 hours for one December day (Fig. B1, Annex B). The winter day (1 December 2018, 12 UTC to 2 December 2018, 12 UTC) was selected so that the synoptic weather situation is comparable to the summer day shown above (weather pattern W1 classification after Irvine et. al, 2013).



**Figure 2: Characteristic patterns of (a) water vapour aCCF [K/kg(fuel)] (b) NOₓ aCCF (including O₃, CH₄ and PMO) [K/kg(fuel)]** **(c) contrail aCCF [K/kg(fuel)] (d) merged non-CO₂ aCCF [K/kg(fuel)] at pressure level 250 hPa over European region for timesteps during 14 June 2018, 12UTC and 15 June 2018, 12UTC (every 6 hours). Individual aCCFs were calculated from high resolution ERA5 reanalysis data.**

We further analyze the distribution of the individual aCCF estimates over the European region at pressure level 250 hPa. Fig.
3 shows the probability density function (PDF) of water vapour aCCFs (Fig. 3a), NOₓ -induced aCCFs (Fig. 3b) and contrail aCCFs (Fig. 3c) for 5 timesteps from 14 June 2018 (12UTC) to 15 June 2018 (12UTC). Over the selected European region (30°-60°N, 25°E-30°W), the water vapour aCCFs show considerable spread (values range between 2.0 $10^{-15}$ and 11.0 $10^{-15}$ K/kg(fuel)) with the highest probability at low aCCF values. However, there is only a small variation in the distribution of



water vapour aCCF over the time steps evaluated, here. In case of the $NO_x$-induced aCCFs the distribution has two separated
peaks and values range from $4.9 \cdot 10^{-14}$ to $5.9 \cdot 10^{-14}$ K/kg(fuel). The different time steps show a quite similar distribution in the
$NO_x$-aCCFs, with some temporal variation (compare e.g. sequenced time steps: 14 June 2018 (12 UTC) and 14 June 2018 (18
UTC)). The PDF of contrail aCCFs over Europe only considers non-zero values, in order to ensure that contrail-free regions
(no climate effect) are excluded in the PDF. The contrail aCCF distributions range from negative ($-4.0 \cdot 10^{-13}$K/kg(fuel)) to
positive ($+4.0 \cdot 10^{-13}$ K/kg(fuel)) values. Time steps during day-time conditions show contrail aCCF distributions, that reveal a
large probability at rather high positive values (between $2.0 \cdot 10^{-13}$ and $3.0 \cdot 10^{13}$ K/kg(fuel)). Overall the distributions have larger
probability for positive values (even during day-time conditions) and variation between the different time steps is larger than
for $NO_x$-induced aCCFs. The shape of the distribution differs significantly between day and night-time (black line) contrail
aCCFs with a narrow distribution during night (0 UTC) with only positive values, that has its highest probability at about $+4.0$
$10^{-13}$ K/kg(fuel). Moreover, the distribution varies within the 24 hours shown, with time steps showing a double peak (i.e. 18
UTC and 12 UTC), and all other time steps showing only one peak.

Additionally, to the spatially and temporally varying aCCFs we include the magnitude of the constant value of the $CO_2$ aCCF
in the water vapour aCCF PDF (Fig. 3a). We see that the $CO_2$ aCCF is for that forward looking metric of F-ATR20 in the
range of water vapour aCCFs, while it is lower than $NO_x$, and contrail aCCF values.

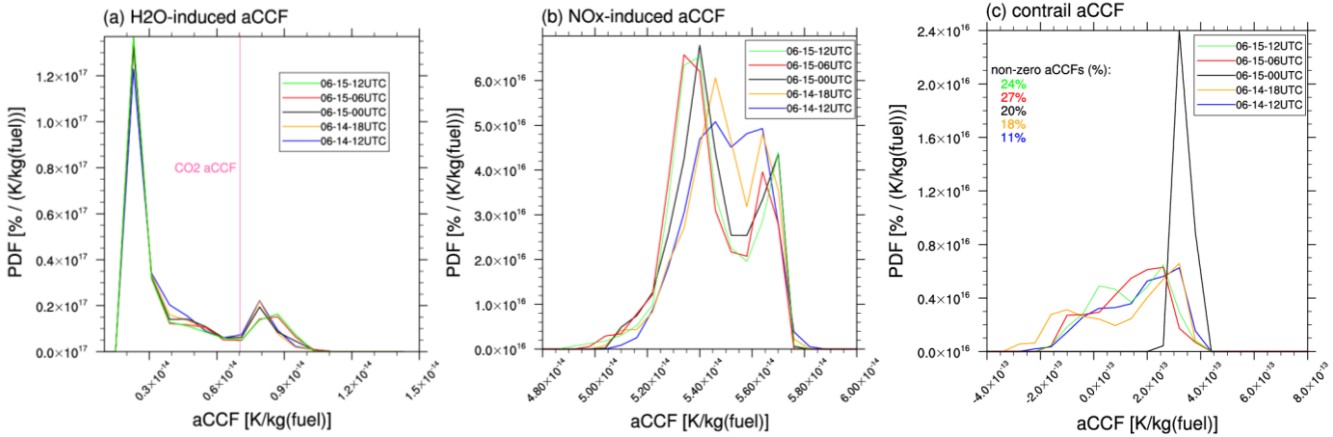

**Figure 3: Probability density function over Europe (30°-60°N, 25°E-30°W) of (a) water vapour aCCF, (b) $NO_x$-induced aCCF, and
(c) contrail aCCF for five different time steps (colored lines) during 14 June 2018 (12 UTC) and 15 June 2018 (12 UTC) at 250hPa.
The PDF of contrail aCCFs is excluding contrail free regions. Percentages of grid boxes with non-zero contrail aCCFs are provided
within (c). The $CO_2$ aCCF, a constant value, indicated within (a).**

Fig. 4 provides additional information on the altitude dependence of the individual aCCFs (i.e., water vapour aCCF (Fig. 4a)),
total $NO_x$ aCCF (Fig. 4b) and day-time contrail aCCF (Fig. 4c) over the European airspace for 15 June 2018 (0UTC) at pressure
levels 200 hPa, 250 hPa and 300 hPa (typical cruise altitudes). All individual non-$CO_2$ aCCF distributions over Europe vary
considerably for these pressure levels. Water vapour aCCFs are higher at high altitudes (200 hPa), as water vapour emitted in
the stratosphere has a high climate effect, because of reduced loss processes of water vapour and its resulting longer lifetimes
(e.g. Rosanka et al., 2020, Frömming et al., 2021). The $NO_x$ aCCF distributions are strongly shifted towards higher values for





pressure level 200 hPa, thus indicating an increase in the climate effect of $NO_x$ emissions for higher altitudes. This increase can be explained by ozone production increase with altitude (sunlight) and reduced ozone loss processes. For the day-time contrails shown the climate effect increases within the altitude range 200-300hPa, indicating that contrails have a higher positive (warming) climate effect at higher altitudes. However, recall that for the contrail aCCFs distribution zero values are excluded and thus the PDF is calculated with a different share of points (respective percentages of grid boxes with contrail

aCCFs are given within the plot).

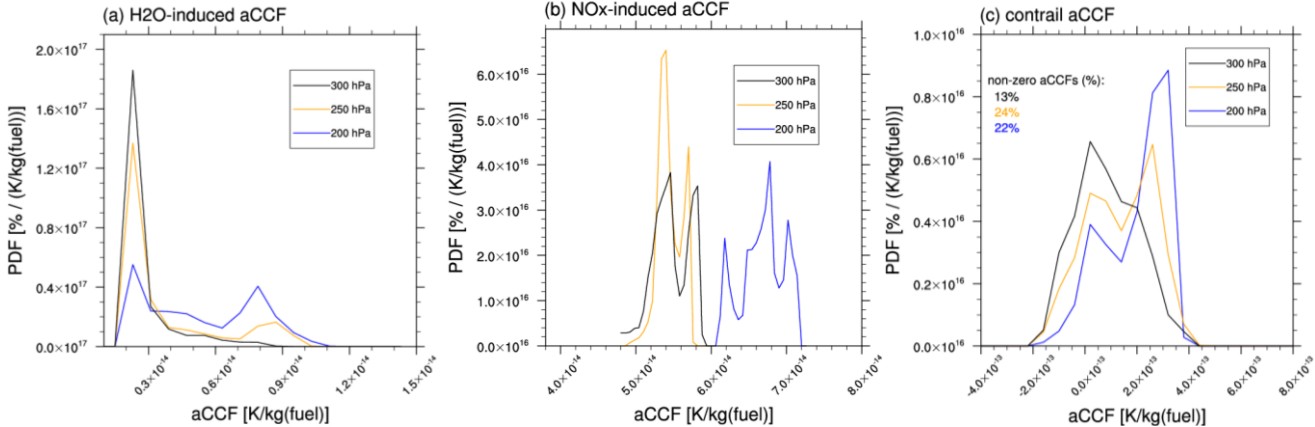

**Figure 4: Probability density function over Europe (30°-60°N, 25°E-30°W) of (a) water vapour aCCF, (b) $NO_x$-induced aCCF, and (c) day-time contrail aCCF for the three different altitudes (200 hPa, 250 hPa, 300 hPa) for 15 June 2018 (12UTC). The PDF of contrail aCCFs is excluding contrail free regions. Percentages of grid boxes with non-zero contrail aCCFs are provided within (c).**

To compare aCCF-V1.0 to aCCF-V1.0A, we show the probability density functions of merged non-$CO_2$ aCCFs (Fig. 5). As the individual aCCFs were calibrated with specific factors ($f_{AirClim}$, Table 2), we expect to observe for the merged aCCFs not only a shift by some factor (as for the individual aCCFs), but also a change in the distribution, this would result in a change in the routing. Both merged aCCF distributions in Fig. 5 show a high relative contribution of those regions where no contrails occur. Note, that in contrast to the contrail aCCF distribution (Fig. 3c) the merged aCCFs include all grid boxes, also contrail-

free regions. Values of the merged aCCFs are for contrail free regions around $0.2 \cdot 10^{-12}$ K/kg(fuel) for V1.0 and $0.05 \cdot 10^{-12}$ K/kg(fuel) for V1.0A, indicating the CCFs of $H_2O$ and $NO_x$. Overall, the merged non-$CO_2$ aCCF-V1.0A are reduced by about a factor of 3 compared to V1.0, while the distribution is narrower.

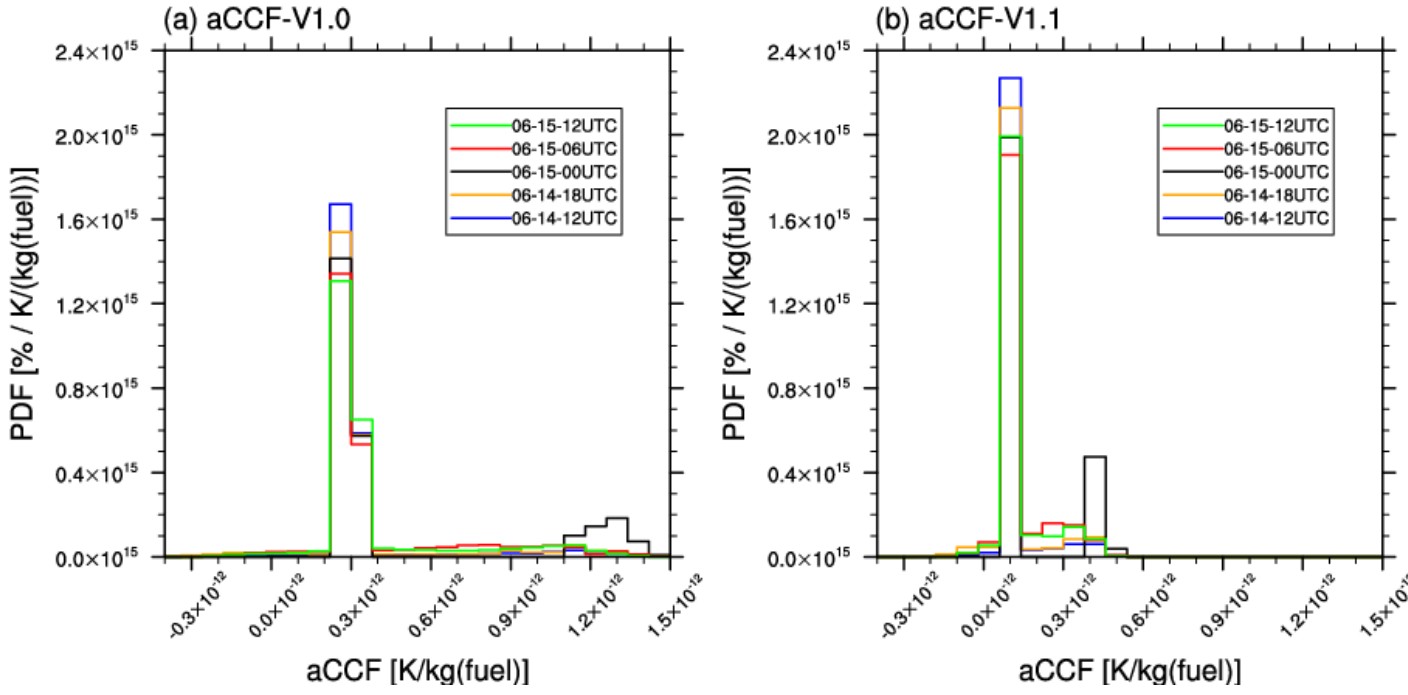

**Figure 5: Probability density function over Europe (30°-60°N, 25°E-30°W, 250hPa) of merged non-CO₂ aCCF for five different time steps (colored lines) during 14 June 2018 (12 UTC) and 15 June 2018 (12 UTC), using (a) aCCF-V1.0 and (b) aCCF-V1.0A.**

## 5. Discussion

In this study we introduce an updated version of aCCFs (aCCF-V1.0A), describing spatially and temporally resolved specific climate effects of aviation emissions, which have been calibrated to estimates of the state-of-the art climate-response model AirClim. Our update is based on the previously published version aCCF-V1.0 (Yin et al., 2022), additionally results from the well-established climate-response model AirClim (Grewe and Stenke, 2008; Dahlmann et al., 2016) are considered, in order to explore the current level of scientific understanding of individual climate effects of aviation emissions. By introducing AirClim calibration factors, we align the results of the aCCFs model with the state-of-the-art scientific understanding regarding individual non-CO₂ effects.

However, it has to be noted here, that the experiment design of our calibration procedure influences the calibration factors derived. Specifically, the calibration procedure applied in this study is based on a European air traffic sample. We restrict the air traffic sample to city-pairs in Europe, as the geographic region where aCCFs have been derived is in particular for the North Atlantic Flight Corridor (together with parts of USA and Europe). Applying these aCCFs to other regions of the globe, would introduce additional uncertainties requiring an evaluation of archived quantitative estimates. However, as air traffic continuously grows in many regions of the globe (e.g., Asia Pacific), there is a strong need to expand the aCCFs for global



coverage. Such expansion is suggested to be addressed in future research work requiring the application of comprehensive global chemistry-climate modelling. First steps were already undertaken at least for $NO_x$-ozone responses (Maruhashi et al, 2022). For the air traffic sample, the calculation of the associated emission inventory, as performed in the chemistry-climate

model EMAC/AirTraf/ACCF, assumes trajectory planning on great circles with a fixed cruise flight altitude of FL350, typical cruise altitude. However, calibration factors might change when trajectories are distributed over different geographic regions, seasons, and cruise altitudes, or if they are optimized to reduce, e.g. operational costs, fuel usage or climate effects. Alternative routing strategies are expected to introduce flight altitude changes, since aircraft performance and non-$CO_2$ effects and hence individual response functions are characterized by strong geographical and vertical dependence.

With regard to the physical climate metric of ATR20 that was selected for the calibration process, we would like to mention, that ATR20 causes the relative importance of aviation $CO_2$ climate effects to be low, compared with aviation effective radiative forcing (ERF) estimates involving historic emissions (e.g. Lee et al., 2021). Here, we apply a forward-looking climate metric considering current emissions (P-ATR20). The underlying question is "How much is the rerouting addressing the global mean temperature change in support of the Paris Agreement in the short-term". Following Grewe and Dahlmann (2015), this question

very much defines the climate metric, consisting of an emission (here pulse emission), climate indicator (here: averaged temperature response to address the overall target of the Paris Agreement), and time horizon (here: 20 years, in order to address the possibility of climate optimized routing as short-term mitigation option in terms of implementation and effect). An alternative could be the F-ATR20 or F-ATR100 climate metrics that address the usage of climate optimised routing on a routinely basis, which is represented in the emission by a future scenario (F) instead of a pulse (P) (see e.g. Grewe et al. 2014)

and a different perspective from short-term to long-term climate impacts (100 years, instead of a time horizon of 20 years). Here the ratio of non-$CO_2$ to $CO_2$ is 8.9 based on Table 2, for a 100-year time horizon (P-ATR100) it would reduce to 4.95. This value can be compared with the GWP* value of 3.0 given in Lee et al. 2021 considering that we consider here a more modern aircraft compared to a whole fleet mix in Lee et al. that also includes older aircraft with a larger fuel use and thereby $CO_2$ emissions.

The difference between the 20- and 100-year time horizon results from a different weighting of individual effects. Due to the chemical and physical processes, short-lived aviation-induced climate effects, like ozone and contrails, have a large RF in the year of the emissions, while the RF associated with $CO_2$ emissions, increases over time due to its long atmospheric lifetime. Analysing solely pulse or future emissions, without considering historical emissions reduces the relative importance of the climate effect associated with $CO_2$ emissions. As a consequence, the aviation $CO_2$ climate effect contributes only less than

10% to the total climate effect, which is composed of $CO_2$ and non-$CO_2$ effects.

One further limitation in the comparison and calibration of climate effect estimates, is the different base years underlying the calculations in the two distinct models. AirClim relies on data extracted from a free running chemistry-climate model simulation, not nudged to reanalysis data, while greenhouse gas concentration and sea surface temperature were representative for conditions of the year 2000 conditions. In contrast, the respective formulation of aCCFs has been derived from base

simulations for the year 2010, which are then applied in the subsequent step to EMAC/AirTraf/ACCF nudged to reanalysis



data from the year 2016 (as discussed below). Here, the different base years used in the construction of the response surfaces introduce an inconsistency, with AirClim referring to the year 2000 conditions, e.g. with lower $CO_2$ concentrations resulting in lower atmospheric temperatures. Hence, the usage of these different meteorological conditions introduces additional uncertainties, assuming year 2000 in the one model and 2010 conditions in the other one.

Overall, we would like to stress that the AirClim calibration factors (Table 2), which calibrate non-$CO_2$ effect estimates from EMAC/AirTraf/ACCF to the values quantified by AirClim, result in confidence intervals for effective radiative forcing of individual climate effects which agree with those suggested by assessment studies, e.g., Lee et al., 2021. In our calibration process, those non-$CO_2$ effects are included where advanced scientific understanding and knowledge is available in order to have correlations available which allow to generate and establish response functions. However, from a conceptual point of

view, the aCCF concept shown in this study can be applied to any other non-$CO_2$ effects, e.g. for aerosol-cloud-interaction once quantitative estimates and improved understanding of atmospheric processes are available (see e.g. Righi et al. 2021).

Despite the updated aCCFs resulting from the calibration procedure presented here, it needs to be stressed that the development of aCCFs remains a continuous work process requiring further updates in the future. A highly parameterized approach is required in order to develop these response functions and surfaces due to assumptions and chosen representation of physical

and chemical processes for such atmospheric comprehensive modelling. The prevailing limitations in the representation of atmospheric conditions and associated uncertainties, e.g. model resolution and cold bias, enter directly into the calibration process. E.g., Yin et al. (2022) discussed several open issues to be addressed further, e.g., the temporal and spatial applicability of the current prototype aCCFs and the uncertainties related to contrail aCCFs. Additionally, initial studies have been undertaken to verify that the aCCF provide climate effect information in an efficient way, which leads to a reduction in the

overall aviation effect as calculated in a global chemistry-climate model, if aircraft fly on optimized trajectories considering aCCF information, e.g. verifying the $NO_x$-induced effects (Yin et al. 2018, Rao et al., 2022).

From a methodological point of view, we **explore in our calibration workflow a novel combination** of response functions which have been established by two distinct, conceptually different approaches regarding two aspects: (a) climatologically-

averaged atmospheric responses that are combined with the emission data versus synoptically-resolved atmospheric responses that are first combined with emission data and then climatologically-averaged, and (b) tagging versus perturbation approach. On the time integration aspect (a) the climate-response model AirClim quantifies effects on an annual mean basis as a mean response of all occurring weather situations, while the conceptual difference of aCCFs is that their quantification is dependent on the individual weather situation. In our evaluation we apply the aCCFs to quantify climate effects as a temporally resolved

effect (with a time resolution of the model time step in EMAC). In order to calculate the total effect of this traffic sample for one year, we integrate over the analysis period. In contrast to this, AirClim directly provides an annual total effect, assuming that the aircraft operations are performed daily during this one year. The conceptual difference is that in one case, integration of effects is done before deriving the atmospheric response functions to emissions (AirClim), while in the other case, the integration is performed applying weather (synoptic) dependent response functions (aCCFs) over one year, and subsequently





aggregating the effects. Hence, the approach followed in this paper (which has been previously done in Yin et al. 2023)
combines an evaluation of effects as an annual mean with an integration of temporally resolved effects, representing two
different model philosophies. This conceptual difference causes additional uncertainties: first, usage of temporal averages
causes a loss of certain information, but also a reduction of extreme values. Second, by applying aCCFs to atmospheric data
for a given point in time, the evaluation is specifically performed for the state of the atmosphere representative for the
respective base year (e.g. 2016), resulting in additional uncertainties being introduced in our evaluation and calibration. As
consequence from our approach, we need to mention here that, despite knowing that the aCCFs have been derived from a small
number of sample days in winter and summer, we extend their application also to other dates and seasons as we compare an
annual mean climate effect from aCCF application with the annual mean AirClim estimate.

On the contribution methodology applied in the simulations (b), we would like to mention that two different approaches when
calculating contributions of individual emissions to climate effects were applied in the two models. While the numerical
simulation concept for the climate-response model AirClim is based on a perturbation approach (Grewe and Stenke, 2008,
Dahlmann et al., 2016), the construction of CCFs (and thus aCCFs) is based on a tagging approach (Grewe, 2013, Grewe et
al. 2017). Note that, in general, climate assessment studies rely on comprehensive perturbation studies of individual aviation
non-$CO_2$ effects (e.g. Lee et al., 2021, Matthes et al. 2021). Earlier studies showed, that tagging estimates can result in higher
estimates on climate effects (Grewe et al., 2012), as a non-linear behaviour of the atmospheric response, e.g. due to non-linear
photochemistry, is shared between individual source sectors and not attributed to a single sector, e.g. the one that is added as
last contribution (as done in the perturbation approach). However, as currently available literature has presented relationships
between tagging and perturbation approach only for annual mean emissions, we decided not to scale any of the quantitative
estimates because such a relationship has not yet been systematically described for individual synoptic situations.

Finally, we would like to draw attention to the **practical implications of the usage of the aCCFs**. First, the aCCF development
is based on EMAC simulations, using a horizontal resolution of T42 (corresponding to a quadratic Gaussian grid of ∼ 2.8 ×
2,8) and 41 vertical layers from the surface to 5 hPa for 8 individual days (5 winter days, 3 summer days) (for details on the
model setup, see Frömming et al. 2021). However, due to a lack of alternative versions of aCCFs, users tend to apply aCCF
also to meteorological data with different horizontal resolutions, e.g. to ERA5 high resolution realization data set with a higher
horizontal resolution (i.e. 0.25° x 0.25°). We clearly want to state here, that applying aCCF developed for T42 to a different
(higher) resolution brings additional uncertainties to the calculated aCCFs. In an application study of aCCFs for trajectory
optimization, we performed an initial evaluation, by comparing aCCF calculated with EMAC input data (T42) to aCCF
calculated by ERA5 data (0.25°).  E.g. in case of ozone the comparison showed that values agreed on average within 10%.
However, it needs to be stressed here, that application of higher resolution always introduces from a numerical diffusion point
of view higher extreme values and low resolutions always cause a reduction of extreme values. Conceptually speaking, one
possible way to account for this influence could be to introduce resolution dependent aCCFs in future studies. Similarly,
beyond the integration of the algorithms in the overall workflow, one critical aspect during the implementation of the contrail
aCCF is the description of ice supersaturation within the respective atmospheric model. The contrail aCCFs only provide non-





zero values for those regions where contrail formation is possible. However, due to the discrete (coarse) horizontal resolution
of atmospheric models, it is not straightforward to determine these threshold values for contrail formation from meteorological
data. We thoroughly evaluated the threshold values in the geographic focus area (Europe and North Atlantic Flight Corridor)
of aCCF models. The idea is to ensure that the values used would lead to a probability of ice supersaturated regions (ISSR)
which agrees with observational data from the European research programme MOZAIC and IAGOS (Petzold et al., 2020).
The detailed process has been documented by Dietmüller et al. (2022), where a value of, e.g. 0.9 has been derived for ERA5
high resolution (0.25°) data.

A point we also want to note is that alternative methods for assessing spatially and temporally resolved information of aviation
induced climate effects are published (e.g Köhler et al. 2008, Rädel and Shine, 2008; Grewe et al., 2010). Regarding the
contrail effect the Contrail Cirrus Prediction model CoCiP (Schumann, 2012; Schumann et al., 2012) can be used to investigate
and predict contrail occurrence, contrail properties and the contrail climate effect. In its updated version (Teoh et al. 2020,
2022) CoCiP calculates the climate effect of single contrails, based on current weather predictions especially of temperature
and humidity. It can be applied on a regional and global scale and allows to determine the expected climate effect due to
persistent contrails for every single flight in the given airspace. A detailed comparison of CoCiP and contrail aCCF estimates
would require a systematic analysis, as both methods rely on different architectures on how such type of information is
implemented (e.g. relying only on meteorology at the location of emission, or relying on the full meteorological forecast).

Additionally, for the calculation of contrail climate effects, we note here that depending on the model the distance of contrail
formation can be calculated with two distinct approaches. In general circulation models, e.g. in EMAC/AirTraf the flight
distance where contrails form is calculated by multiplying the flown kilometres in a model grid box with the fractional potential
contrail cirrus coverage (a value between 0 and 1). Alternatively, the application of aCCFs can rely on the total flown kilometre
in the respective geographical grid box as the required input information. This implementation is valid for high resolution
numerical weather forecast data, when assuming that the individual grid boxes are small enough to assume either 0% or 100%
contrail cirrus cloud cover.

## 6 Conclusion & Outlook

This study presents aCCF-V1.0A, an updated formulation of aCCF-V1.0, which is calibrated to a recent version of the climate-
response-model AirClim. The evaluation of non-$CO_2$ climate effects from an emission inventory of European city pairs
performed in this study has been the basis for the formulation of the updated aCCF-V1.0A. Our analysis illustrates to what
extent current the formulation of aCCFs is still subject to large uncertainties, shown by the AirClim calibration factors of up
to an order of magnitude. As a result, from the calibration performed the updated formulation V1.0A is now available enabling
an estimation of the climate effects of aviation representing the relative strength of the non-$CO_2$ effects among each other,
aligned with state-of-the-art literature.





However, the formulation presented in this study, still relies on the initially available comprehensive modelling results for the North Atlantic Flight corridor. In order to further develop the strength and capabilities of the aCCFs concept, comprehensive atmospheric chemistry climate modelling is required relying on state-of-the-art chemistry-climate models, expanding the geographic regions where such comprehensive Lagrangian assessments are performed. In such simulation setups the quantification of the radiative effects plays an important role and requires particular attention. Such further knowledge can

then constitute the basis for a further update of the aCCF formulations. In particular an extension of this aCCF concept is required for other regions of the world, as well as other seasons. Additionally, with the help of model evaluation it will be key to on the one hand further reduction of uncertainties, but on the other hand, the development of methods and concepts on how to deal with uncertainties in a robust decision-making process.

**Data and code availability**

ACCF and AirTraf are implemented as submodels of the Modular Earth Submodel System (MESSy). MESSy is being continuously developed and applied by a consortium of institutions. The usage of MESSy and access to the source code are licensed to all affiliates of institutions which are members of the MESSy Consortium. . Institutions can become a member of the MESSy Consortium by signing the MESSy Memorandum of Understanding. More information can be found on the MESSy Consortium website (https://messy-interface.org/). The software code AirClim is confidential proprietary information of DLR.

Therefore, the code cannot be made available to the public or the readers without any restrictions. Licensing of the code to third parties is conditioned upon the prior conclusion of a licensing agreement with DLR as licensor.. The emission inventory calculated by EMAC/AirTraf which has been evaluated by AirClim in this paper will be provided by the authors on request. The used ERA5 reanalysis (Hersbach et al., 2020) data are available at the Copernicus Climate Change Service Climate Data Store via https://doi.org/10.24381/cds.bd0915c6.

**Authors contribution**

SM prepared the manuscript with contributions from all co-authors. SM, SD and FY designed the calibration concept; SM, SD, KD, CF, PP, HY, VG, FY and FC wrote the manuscript draft; SD, CF and PP described the atmospheric state; HY, FC and FY performed the trajectories simulation with EMAC/AirTraf; KD performed the AirClim simulation; FY reviewed and edited the manuscript.

**Competing interests**

At least one of the (co-)authors is a member of the editorial board of Geoscientific Model Development.

**Acknowledgments**

The current study has been supported by FlyATM4E project, which has received funding from the SESAR Joint Undertaking under grant agreement No 891317 under European Union's Horizon 2020 research and innovation program. Moreover, this project has received funding from the SESAR Joint Undertaking (JU) under grant agreement No 891467. The JU receives support from the European Union's Horizon 2020 research and innovation program and the SESAR JU members other than the Union. Additionally, the ALARM project (SESAR JU under grant agreement No 891467), coordinated by Manuel Soler (UC3M Madrid) supported this study. High performance computing simulations with the chemistry-climate model EMAC were performed at the Deutsches Klima-Rechenzentrum (DKRZ), Hamburg, and by the TU Delft High Performance Cluster (HPC12).

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

**ANNEX A: Mathematical formulation of aCCF-V1.0A**

**Ozone aCCF**

The mathematical formulation for the ozone aCCFs is based on temperature $T$ [K] and geopotential $\Phi$ [m$^2$/s$^2$]. The relation for the ozone aCCFs (aCCF$_{O3}$) at a specific atmospheric location and time is given in temperature change per emitted NO$_2$ emission [K/kg(NO$_2$)]:

$$aCCF_{O3} = \left\{ \begin{array}{ll} aCCF'(T,\Phi) & \text{for } \begin{array}{l} aCCF' > 0 \\ aCCF' \leq 0 \end{array} \\ 0 & \end{array} \right.$$

with $aCCF'(T,\Phi) = f_{AirClim}(O3) * (-2.64 * 10^{-11} + 1.17 * 10^{-13} * T + 2.46 * 10^{-16} * \Phi - 1.04 * 10^{-18} * T * \Phi)$

Accordingly, the ozone aCCFs takes positive values, and is set to 0 in case of negative aCCF' values.

**Methane aCCF**

The methane aCCFs is based on the geopotential $\Phi$ [m$^2$/s$^2$] and the incoming solar radiation at the top of the atmosphere $F_{in}$ [W/m$^2$]. The relation of the methane aCCFs (aCCF$_{CH4}$) at a specific location and time is given in temperature change per emitted NO$_2$ emission[K/kg(NO$_2$)]:

$$aCCF_{CH_4}(\Phi, F_{in}) = \left\{ \begin{array}{ll} aCCF'(\Phi,F_{in}) & \text{for } \begin{array}{l} aCCF'(\Phi,F_{in}) < 0 \\ aCCF'(\Phi,F_{in}) \geq 0 \end{array} \\ 0 & \end{array} \right.$$

with $aCCF'(\Phi, F_{in}) = f_{AirClim} * (-4.84 * 10^{-13} + 9.79 * 10^{-19} * \Phi - 3.11 * 10^{-16} * F_{in} + 3.01 * 10^{-21} * \Phi * F_{in})$

Thus, the methane aCCFs is negatively defined. It is set to 0 if the term aCCF' is 0 or positive.



$F_{in}$ is defined as incoming solar radiation at the top of the atmosphere as a maximum value over all longitudes and is calculated by: $F_{in} = S * cos\theta$ , with total solar irradiance S=1360 Wm$^{-2}$ , with $cos\theta = \sin(\varphi)\sin(d) + \cos(\varphi)\cos(d)$

and with $d$ = -23.44° cos(360°/365 * (N + 10)). Here θ is the solar zenith angle, φ is latitude, and $d$ is the declination angle, which defines the time of year via the day of the year N.

The mathematical formulation of the ozone aCCF is only valid for the short-term ozone effect of NO$_x$. The primary mode ozone (PMO), which describes the long-term decrease in the background ozone, as result of a methane decrease. Thus, if merging total NO$_x$ effect be aware that only the NO$_x$ effect on short term ozone increase and on methane decrease is taken into

account. For NO$_x$ induced PMO climate impact we have the possibility to include it to the total NOx aCCF, as the PMO aCCF can be derived by applying a constant factor of 0.29 to the methane aCCF.

**Water vapour aCCF**

The water vapour aCCFs is based on the Potential Vorticity ($PV$) given in standard $PV$ units $[10^{-6}\text{K kg}^{-1}\text{ m}^2\text{ s}^{-1}]$. The

following the relation of the water vapour aCCFs (aCCF$_{H20}$) at a specific location and time is given in temperature change per fuel [K/kg(fuel)]:

$$aCCF_{H_2O}(PV) = \text{f}_{\text{AirClim}}(\text{H2O}) * (2.11 * 10^{-16} + 7.70 * 10^{-17} * |PV|)$$

The absolute value is the $PV$ is taken to enable a calculation on the southern hemisphere, where $PV$ has a negative sign.

**Contrail aCCF**

The aCCFs of persistent contrail cirrus have been developed within the EU project ATM4E. A detailed description and verification of these contrail aCCFs are given in the supplement of Yin et al, 2022.

Contrail aCCFs are calculated separately for day-time and night-time contrails, because their climate impact differs between daylight and darkness, as the shortwave forcing is only relevant for daylight conditions. To differ between day-time and night-

time contrail aCCFs, the local time and solar zenith angle are calculated. For locations in darkness, the time of sunrise is calculated. If the time between the local time and sunrise is greater than 6 h, the night-time contrail aCCFs is applied. In order to determine the contrail aCCFs, the RF of day-time or night-time contrails is calculated as described in the following.

The **RF of day-time contrails** (RF$_{\text{aCCF}-\text{day}}$) in [W/m$^2$] is based on the outgoing longwave radiation ($OLR$) in [W/m$^2$] both

at the time and location of the contrail formation. For a specific atmospheric location and time, the RF$_{\text{aCCF}-\text{day}}$ is given by:

$$RF_{aCCF-day}(OLR) = 10^{-10} * (-1.7 - 0.0088 * OLR)$$

According to the equation, the RF for the daytime contrails can take positive and negative values, depending on the $OLR$.

The **RF of night-time contrails** (RF$_{\text{aCCF}-\text{night}}$) in [W/m$^2$] is based on temperature ($T$) in [K]. For an atmospheric location

(x, y, z) at time t:





$$RF_{aCCF-night} = \left\{ \begin{array}{l} RF_{aCCF-night}(T) = 10^{-10} * (0.0073 \times 10^{0.0107*T} - 1.03) \\ 0 \end{array} \right\rbrack \text{ for } \begin{array}{l} T > 201K \\ \\ else \end{array}$$

For temperatures less than 201 K, the night-time contrail is set to zero.

The above calculated RF of contrails can be converted to global temperature change (ATR20) by just multiplying with a constant factor of 0.0151 K/(W/m$^2$) (Dahlmann, pers. Communication, 10/2021).


$$aCCF_{contrail} = f_{AirClim}(contrail) * 0.0151 * RF$$








**Annex B: Additional plots**

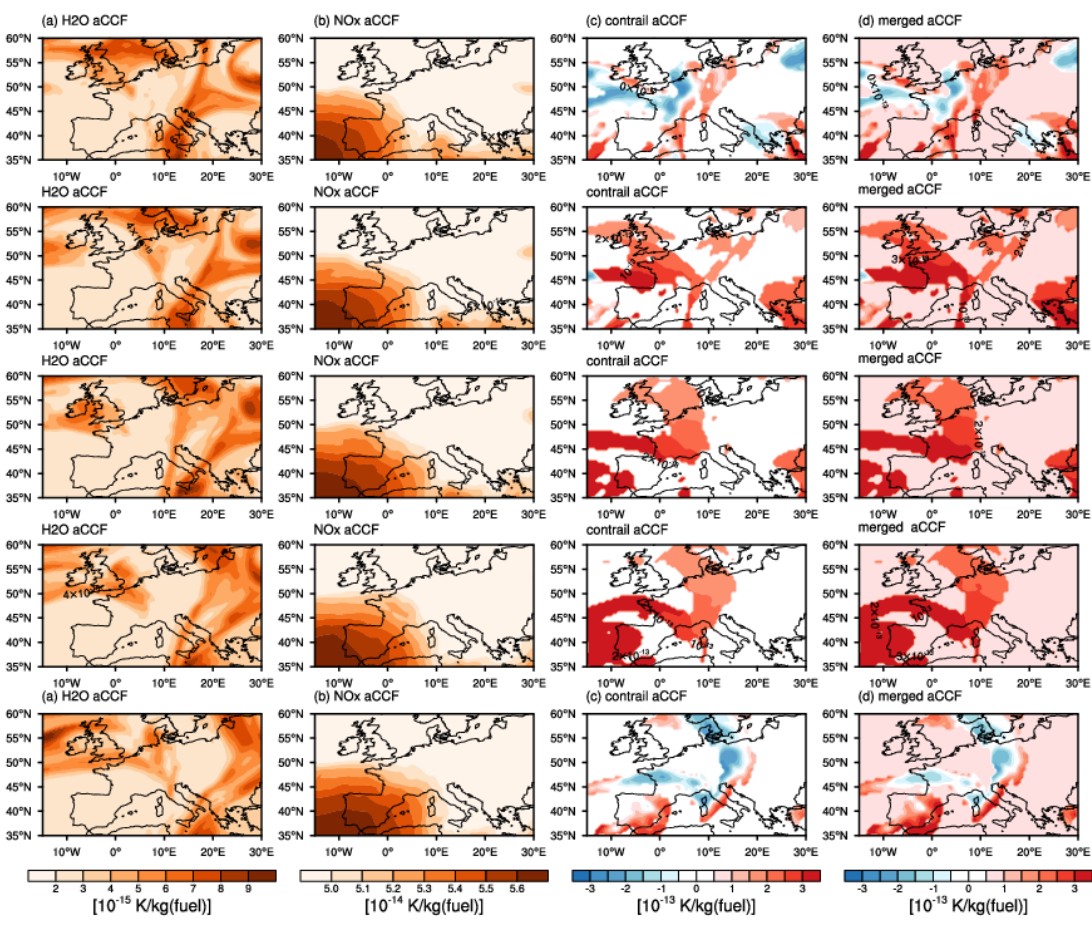

**Figure B1: Characteristic patterns of (a) water vapour aCCF [K/kg(fuel)] (b) NOₓ aCCF (including O₃, CH₄ and PMO) [K/kg(fuel)] (c) contrail aCCF [K/kg(fuel)] (d) merged non-CO₂ aCCF [K/kg(fuel)] at pressure level 250 hPa over European region for timesteps during 1 December (12UTC) and 2 December 2018 (12UTC) every 6 hours. Individual aCCFs were calculated from high resolution ERA5 reanalysis data.**
