# Peer review of "Updated algorithmic climate change functions (aCCF) V1.0A: Evaluation with the climate-response model AirClim V2.0"

_Geoscientific Model Development, 2023_

## Author Comment (AC2)

We thank the reviewer for the constructive and helpful input and provide our replies in detail below.

**REFEREE 2 (RC2)**

In this manuscript the authors compute the non- $CO_2$  effects of aviation using their algorithmic climate change functions (aCCF, V1.0), which themselves are defined from a suite of models, and the AirClim v2.0 model for a set of city-pairs in Europe and a full seasonal cycle. From these simulations they compute renormalization factors (one factor for each non- $CO_2$  effect) to derive calibrated aCCFs (V1.0A). The authors also present PDFs of the aCCFs over a European domain and a comparison between the uncalibrated and calibrated aCCF.

I have a number of major issues with this manuscript. In my opinion each one of these issues could prevent publication in its own right so I urge the authors to consider them carefully.

1/ Calibration implies reference.

Calibration is the process by which a measurement is adjusted to a reference. It is not clear to me why AirClim V2.0 (as described in Dahlmann et al. 2016) provides such a reference. It is claimed that AirClim is "comprehensive", "state-of-the-art", "well-established" but this is not demonstrated. I would welcome elements that prove AirClim is a reference. To which extent is it validated against observations? How is it documented? How does it compare to other models? Dahlmann et al (2016) is now 8 years old, is it still state-of-the-art?

• We agree with the reviewer that a more detailed description of the verification that has been done on AirClim was missing in the text.

[line 181 ff] We added a paragraph at the end of Sect. 2.3 that aims at proving that AirClim shows a) similar RF values compared to previous assessment (Lee et al. 2021) and b) gives sensitivities that are in the range to those calculated by comprehensive climate-chemistry models.

- We stated in line 53 that we think that AirClim is a more "comprehensive" climate-response model than that used by Schwartz-Dallara. Simply because the spatial resolution in AirClim (latitude and altitude dependency and for contrails in addition longitude dependency) and the consideration of contrail-cirrus instead of line-shaped contrails is more advanced.
  [line 56] We change the wording to "advanced".
- As a consequence, we have reconsidered using the term 'calibration' in the manuscript. What we do, is scaling individual non-CO2 effect estimates from aCCFs in a way that they become similar to absolute estimates from AirClim. This results in this updated aCCF version V1.0A which is still showing an identical spatial and temporal "relative distribution" of individual non-CO2-aCCFs-effects, but results in changing relative contributions of individual non-CO2 effect to the overall climate effect. These changes in the relative contributions happen not only in the overall totals, but at any given point where aCCFs are applied.

Hence, we suggest to replace the term 'calibration' in the whole manuscript with 'alignment'.

**2/ Calibration parameters are beyond reasonable.**

The calibration parameters (f\_AirClim in Annex A) are much lower than 1 for some of the non-CO2 effects, i.e. 0.333 for contrail-cirrus, 0.179 for O3 and even 0.058 for CH4. Rather than just rescaling their parametrisation, the authors should go back to their models, understand the root causes of the differences, reformulate the model and try to narrow down the discrepancies. I personally have little trust in the structure of a model that requires a scaling factor of 0.058 against a reference.

• Thank you for the comment. Having alignment factors which are not close to 1 identifies the prevailing challenges in having absolute estimates of climate effects available from numerical tools, which is the case for NOx-induced ozone and methane, as well as contrails. Here, we can note that our alignment factors all go into the same direction (as they are smaller than 1, even smaller than 0.4). Here, these factors deliver an essential information of the interrelationship of individual non-CO2 effects as delivered by these two distinct numerical tools (for a European traffic sample), showing that aCCF estimates are considerably larger. In order to provide further information on some sensitivities of these factors, we propose to add alignment factors from additional simulations for two neighboring flight altitudes (FL330, FL370). We find similar factors also for the other flight altitudes, while from the figure one can identify to what extent the individual tool is providing different estimates for different altitudes (see new table in Annex and new Figure 6).

[line 204f] [..] using three different ceiling heights at flight altitude FL350 (10668 m, 240 hPa), FL330 (10058 m, 260 hPa) and FL370 (11277 m, 215 hPa).

• We want to stress here, that the idea behind alignment of aCCFs to AirClim estimates, is that with this scaling we can maintain the relative spatial and temporal variation of climate effects from aviation, e.g. on the related synoptic scales. However, at the same time, we bring absolute values for European traffic "down" to the values of AirClim. We have added a section in the discussion (5.2 Size of alignment factors), where we discuss this in more detail.

[line 374] 5.2 Size of alignment factors // The alignment factors derived within the present study clearly deviate from 1, specifically being lower than 0.35 for nitrogen oxide induced effects on ozone, methane and contrails, with 0.15, 0.06 and 0.33 respectively.

**3/ Tagging vs perturbation approach.**

As stated on AirClim is based on a perturbation approach (as stated on lines 54-55), i.e. it computes the marginal change in RF due to the perturbation of NOx from aviation emissions. My understanding is that aCCFs V1.0 are based on the tagging approach which places all the NOx emissions (from different sectors) on a same footing. The perturbation (or marginal) approach has the advantage of representing the change expected from an action (e.g. rerouting) but has the disadvantage of not being additive and the sum of the relative perturbations from different sectors or regions usually do not add up to 100%. The tagging (and other similar methods) have the advantage of being invariant to disaggregation and recombination and are a better measure of the contribution of a sector to the total forcing but are not adapted to quantifying the impact of a change when everything else is fixed. My understanding is that the calibration process shifts the aCCF from the tagging to the perturbation approach (line 386). For the methodology to be valid, the authors need to show that there is a proportionality coefficient between the two approaches that holds for individual flights and not just on average. Otherwise there should be a clear warning that aCCFs V1.0A should not be used for individual flights.

• Thank you for the advice. It is correct, that conceptual differences between tagging and perturbation approach exist, which are contributing to the derived alignment factors. Recent work by Maruhashi et al. (under revision) showed that the sensitivities of aviation NOx emissions between the tagging and the perturbation approach are similar. The work also indicates that the principle responses between the perturbation and tagging approaches using EMAC as a climate-chemistry model and the AirTrac-Lagrangian version behave consistently, but not identical. Especially, the two tagging approaches EMAC/TAGGING (full chemistry) and here used EMAC/AirTrac (full chemistry calculated and production rates linearised at every timestep to apply on air parcels) show a systematic difference.

- Results indicate that no constant factor is identified, but variations in the order of 20% can be identified. Hence, we consider that changing from perturbation to tagging approach introduces an additional uncertainty in the order of 20%, but does not affect the overall validity of the aCCF concept.
- With the proposed additional sensitivity study (repeating the workflow for two neighboring flight levels), we derive separate set of alignment factors, while exploring vertical dependence of such differences. We have updated and expanded the discussion on tagging and perturbation approach in the discussion accordingly. Hence, overall we conclude, that aCCFs can be applied to provide estimates on single flights.

[Line 521-526] This is not an issue for  $NO_x$ , only, but applies also to other emissions such as CO2 (Boucher et al. 2021). It should be noted that changing from perturbation to tagging approach introduces differences and additional uncertainties. Maruhashi et al. (2024) investigated those differences in more detail and found that although the differences in the methods are significant (in the order of 20% to 160%) the sensitivities and behaviour in both approaches with variation in altitude and region are similar. Overall, we presume, that an uncertainty of 20% does not affect the overall validity of our approach. However, as currently available literature has presented relationships between tagging and perturbation approach only for annual mean emissions, we decided not to scale any of the quantitative estimates because such a relationship has not yet been systematically described for individual synoptic situations.

**4/ There is a serious lack of treatment of uncertainties for aCCF V1.0A.**

In their present form aCCFs do not come with uncertainties. To be useful, aCCFs should be associated with an uncertainty range (e.g., 1 or 2 sigma or 90% uncertainty range). This manuscript offers the opportunity to address this lasting deficiency of aCCFs by tracking down and combining the different sources of uncertainties. First of all, the aCCFs are expressed as a function of a very limited set of predictive variables and there is a significant dispersion around the average relationship. This implies an uncertainty range that needs to be documented and propagated into aCCFs v1.0A. AirClim V2.0 also has uncertainties that probably add quadratically to those of the aCCF formulation. A third source of uncertainty comes from the representativity of a climatological calibration for individual flights. This can be diagnosed from the dispersion of the calibration coefficients estimated on a flight-by-flight basis and should also be added quadratically to other sources of uncertainties. Finally, there are other sources of uncertainties (e.g., the masking of source regions for the contrail aCCF) that need to be discussed if they cannot be estimated properly.

• We agree with the reviewer that the aCCFs are facing uncertainties from different sources as e.g. meteorological forecast, climate science, or tin he statistical approach used by aCCFs. We aim to provide confidence intervals (lower and upper estimates) of aCCFs values that include all uncertainties that can be quantified. In case of meteorological forecast we can use e.g. ensemble forecast - that was also done in the FlyATM4E project- here the meteorological uncertainty (characterized by employing ensemble forecast) was addressed in trajectory optimizatio (robustness with less sensitivity to meteorological uncertainty) Simorgh et al. 2023. We added a section (5.3) in the discussion section on uncertainties.

[line 398-416] We consider treatment of uncertainties is a central point when using aCCFs for climate-optimization of aircraft trajectories (Matthes et al., 2020). ACCFs are facing uncertainties from different sources which for a more systematic quantitative assessment can be categorized in different groups. One first group of uncertainty could be related to meteorological forecasts and representation of background atmospheric conditions in numerical models. Second, aCCFs deal with uncertainties related to estimating the overall climate effect in Earth-System climate models, e.g., caused by radiative transfer calculation, by the representation of

atmospheric processes, or by the choice of the used physical climate metric. A third group of uncertainties comprises the statistical approach that has been used to develop the aCCFs by correlation analysis (van Manen and Grewe, 2019). Fourth, uncertainties linked to the calculation methods of aircraft-engine emissions could be identified (e.g. DuBois and Paynter, 2006; Jelinek, 2004). The current lack of a comprehensive statistical treatment of uncertainties in the aCCFs is a challenge that has been pointed out in earlier studies (e.g. Matthes et al. 2020, Dietmüller et al. 2023, Yin et al. 2023, Simorgh et al. 2024; Zengerling et al. 2023). Initial results on such a systematic assessment have been made available from the EU project FlyATM4E (with a publicly available deliverable footnote: https://www.flyatm4e.eu/ deliverables}) and a publication is in preparation, which describes and quantifies uncertainties in detail for a given use case (Matthes et al. 2024). Quantification methods to estimate individual CO2 and non-CO2 climate effects; refer to a mean value, while from a statistical point of view, each estimate is linked to a confidence interval, with each value in this interval being possible. Hence, instead of choosing the mean values to calculate the climate effect, it is equivalent to select another value from this interval. Consequently, uncertainty assessments should lead to identification of a confidence interval for the quantitative climate effect estimates which can be determined relying on error propagation and statistical methods to mathematically describe these identified underlying uncertainties and confidence intervals.

**5/ aCCFs V1.0A lack traceability and transparency**

aCCFs lack the traceability and transparency needed by users. aCCFs V1.0A make the situation even worse as they result from a long suite of models (some of which are not freely available) and simulations (some of which not accessible). I am worried that aCCFs V1.0A will be used without any consideration of their inherent uncertainties and/or outside their validity range. The authors need to provide a more traceable workflow with the input and output of the models that are used to produce the aCCF V1.0A. Only in this way can the aCCF be compared to other approaches and become a trustworthy source of information for users.

• Thanks for the comment. We have updated the description of the workflow (Section 2.1) accordingly, now providing a more detailed description. Specifically, we explain in more detail how the inventory is constructed (EMAC/AirTraf output) and then is used as input for AirClim (and for EMAC/AirTraf/ACCF) in order to estimate annual CO2 and non-CO2 climate effects. Accordingly. we have updated figure 1, now providing more details on emission inventory, e.g. species included, annual basis, and input and output information.

[line 137-144] During a one-year simulation (EMAC/AirTraf/ACCF), individual climate effects are calculated for each time step, and finally summed up in order to calculate the total annual climate effect, comprising CO2 and non-CO2 effects, namely the emissions totals of fuel and CO2 emission and the climate effects of CO2, NOx, water vapour and contrail effects. AdditionallyAs a secondary product EMAC/AirTraf/ACCF outputs constructs an gridded annual emission inventory for this traffic sample, which is subsequently used as an input to AirClim (Dahlmann et al., 2016). Consistently, the climate-response model AirClim calculates the climate effect for the same European air traffic sampleof this inventory assuming constant emissions on an annual basis (Figure 1). By comparing the both estimates on an annual basis of total annual climate effects from EMAC/AirTraf/ACCF and AirClim for each individual specie, a set of scaling factors for NOx, water vapour and contrail effects (i.e., AirClim alignment calibration-factors) is identified, which are then applied to aCCF-V1.0, to construct aCCF-V1.0A.

• Instead of mentioning a personnel communication we have added a recent peer-reviewed manuscript.

• [Line 915] (Yin et al. 2023<del>Dahlmann, pers. Communication, 10/2021</del>). And consequently, to the contrail aCCF-V1.0A is defined as: [formula] Note moreover, that contrail aCCFs are only calculated at location and time where persistent contrails are forming, and accordingly regions without persistent contrails are set to zero.

**6/ The choice of metric is unusual and misleading.**

The default version of aCCFs comes are for the ATR-20 climate metric (temperature change averaged over a 20 year period) and aCCFs V1.0A are also illustrated for the ATR-20. This is an unusual choice within the large body of literature on climate metrics. It also contrasts with policy choices made in UNFCCC (see also recent SBSTA decision to retain GWP100 in the wake of the Paris Agreement). The ATR-20 metric puts a lot of weight on the climate effects of short-lived species. Sure there is a policy dimension in climate metrics but there are also scientific and economical considerations that favour putting more weight on long-lived greenhouse gases, at least until CO2 emissions are curbed significantly. What is so special to the aviation sector to favour ATR20 over metrics with longer time horizon (e.g. GWP100 or GTP50) that are more consistent with the current mitigation levels? At the very least aCCFs V1.0A should be presented with several climate metrics. Highlighting ATR20 can be very misleading for users who are not very knowledgeable on climate metrics.

We agree, that the ATR-20 climate metric is an unusual choice. This choice can be explained by two aspects: **First**, other commonly used physical climate metrics (comprising GWP, GTP, ATR, but also RF) can be calculated from P-ATR-20 values with the help of conversion factors derived from a climate-response model (AirClim), to be applied individually to each non-CO2effect, as e.g. provided in the Dietmüller et al. 2023-GMD paper (Table 3). E.g. multiplying Pulse-ATR20-values for ozone with the conversion factor of 14.5 results in the corresponding F-ATR20-values, or with 34.1 and 58.3 resulting in the F-ATR-50-values and F-ATR100-value, respectively, assuming a BAU as future emission scenario (see Table 3 in Dietmüller et al., 2023). Second, the ATR-20-metric puts higher weights on estimating the climate effects on shorter time horizons, recognizing that not even the temperature response has (completely) built up in the atmosphere. Hence, using only ATR20 as physical climate metrics ignores long-term effects, while on the other hand focusing strongly on evaluating 'quick wins'. These ideas for sectors which might have mitigation potentials on short time horizons, have been spelled out in earlier communications (e.g. Penner et al., 2010, https://www.nature.com/articles/ngeo932). In order to stress this characteristic of ATR-20-metric, we suggest to explicitly mention the fact that by averaging the change of surface temperature over a time horizon of the next 20 years, the temperature response of the atmosphere to the radiative imbalance would only be integrated to a small part. The reason here is that typical response times of the (surface) temperature due to the high inertia of the Earth-Atmosphere system go beyond 20 years, more towards 50-70 years. We suggest to mention this aspect in the introduction and probably in the discussion. [line 216-217] We note that the short time horizon chosen here, puts more weight on the short-term effects, and does not integrate over the whole atmospheric response. [line 434-459] [We further added a separate section on climate metrics in the discussion]

**Other Comments:**

Title: the title does not describe the content of the manuscript. There is no evaluation of aCCF against AirClim V2.0. Performing such an evaluation would imply to show a range of score of aCCF against AirClim V2.0 for individual flights. As mentioned above, I am also dubious that this is a calibration. In any case the title needs to be changed and reflect the content of the manuscript.

*Line 24: this needs to be rephrased as it is not "emissions" per se that are radiatively active or not but the molecules in the atmosphere....*

**• Changed to "non-CO2 effects".**

Lines 27-28: it would be useful to say a bit more about contrails and induced cirrus at this point....

• We added: [Line 31-32] "linear contrails can spread and cause contrail-cirrus. Contrail cirrus can change the radiation budget by longwave (absorption) and shortwave effect (scattering)."

Lines 29-32: the two sentences repeat rather than complement each other. The "accordingly" does not read well. The first sentence omit the dependence on environmental factors (position of the Sun, surface albedo, clouds, ...).

• Repetition deleted (see track changes)

*Lines* 63-64: *see above for an alternative view.*

• We understand from the comment, that the reviewer suggest to further reduce simplifications and parameterisations for providing estimates on climate effects of aviation at a given location and time during a flight planning procedure. As explained above, we recognize that the concepts and methodologies which exist today for providing such estimates, all rely on parameterisations and simplifications, hence we suggest to spell this out also here more specifically. [line 67-68] For this reason, an efficient, **simplified** implementation concept for trajectory planning tools, the algorithmic Climate Change Functions (aCCFs), [..]

Line 69: this may be a strength but this is also their main weakness.

• We added a short sentence on uncertainties in order to explain their relevance for the aCCF concept (see track changes).

Lines 75-76: given the bold assumptions made in aCCF and the absence of characterization of uncertainties, it is dangerous if not fallacious to encourage climate-optimized flight trajectories.

• For further clarification, we have added central characteristics of aCCFs, which are parameterized, simplified and provide estimates of achievable mitigation gains having been explored.

*Lines* 83-84: *I agree that the uncertainty of aCCF estimates is missing but this manuscript does nothing to better quantify such uncertainties.*

• Yes, we agree. We now add a paragraph on uncertainties in the discussion section (using subsection headings for structuring the discussion). See revised document.

*Lines 107-112: this paragraph is not clear and I could not link it to the calibration approach described later in the manuscript.*

• Uncertainty aspects are deleted here and they are now explained in detail in the discussion section (explanation of confidence intervals and range of possible values is moved there). Additionally, we rephrased the description of alignment factors, as being mathematical factors which bring aCCF estimates to AirClim estimates, when multiplying aCCF values. See trak changes in the revised document.

Figure 1: why are the bottom arrows pointing to the green boxes rather than starting from there?

• The arrows have been chosen, to illustrate the alignment of two existing independent estimates. For further clarification, we have updated the figure.

*Line 140: I understand MESSy2 is nudged to ERA5 but why is this considered as "boundary conditions"? What does it imply for the simulation of ISSR? How does it compare to observations?*

• As reanalysis is provided as boundary condition towards which we relax/nudge our model meteorological variables, such nudging field can be seen as boundary condition (no feedback, etc.). However, we have deleted the term boundary condition (see track changes), as it might be confusing. Currently existing reanalysis data are still subject to deficiencies in the upper troposphere when representing temperature and humidity. There are numerous studies which explore these in more detail, e.g. Petzold et al.. As a general rule often they find a cold bias and a wet bias, which is equivalent to an overestimating occurrence of ISSR conditions.

Line 153: see above. Considering great circles is an issue because there are co-variations between the jet stream (which airlines will try to avoid or use depending on the direction) and the presence of ISSR. It is well known that there are significant departures from the great circle. If aCCF are not computed for real trajectories, how can it be assumed that it is useful for designing climate-optimized trajectories?

• When evaluating climate-optimized trajectories, ACCFs are computed along real trajectories. However, in the study presented here it is assumed that aircraft are operated with no dedicated flight guidance strategy, e.g. not wind optimal, not cost-optimal or not climate optimal. Moreover, aircraft are operated on great circles, in order to avoid co-variations between routing strategies and other effects, comprising fuel consumption but also non-CO2 effects encountered. Both numerical tools receive this idealized inventory in order to estimate climate effect of such an "unoptimized" air traffic, which is a central prerequisite for the alignment procedure performed in this study.

*Line 164-165: How does the Green function for CO2 from Hasselmann et al (1997) compare to more recent estimate? To which extent does it depend on future emission scenarios?*

• The Green function from Haselmann et al. (1997) differs from, e.g. Boucher and Reddy (2008). Lee et al. (2021) gave an estimate for the RF in 2018 that ranges between 32 and 39 mW/m2. The respective RF based on the method applied here amounts to 40 mW/m2 (Grewe et al 2021), while utilizing somewhat larger emissions after 2000 compared to Lee et al. 2021 (~10%). Hence the RF calculation is in the range of newer model results. For the relation between RF and temperature change Boucher and Reddy (2008) and Hasselman et al (1997) weight the short- and longterm responses differently. However, what is important for this study is that both models apply identical Green Functions, in order to avoid any influence on the alignment factors here.

*Line 169: is it a "calibration", a "comparison" or an "evaluation" (as per the title)?*

• We understand, that "calibration" might be misinterpreted. We removed the term "calibration" and use the terms "alignment" and "scaling" in the whole manuscript and have adapted the text and the title accordingly.

Table 1: SI units are m and kg rather than km and g. EGU journals recommend to use SI units unless there is a good justification not to. Is there a good reason to deviate from SI units in Table 1?

• We agree and use g as the basic for fuel and emissions. We suggest to deviate from SI-unit for distance flown, as aircraft emission inventories are in general provided as flown kilometer, e.g. https://doi.org/10.5194/acp-24-725-2024. We changed Table 1 accordingly.

Table 1: I assume the 85 flights operate daily during a year, hence the  $\sim$ 50 millions km flown. Unless I missed the information, the fact that the flights are assumed to operate daily is missing.

• Thank you, we have added this information to the caption of Table 1, "operating on a daily basis".

Line 229: Kg should read kg.

• Done.

*Line 229 and elsewhere: for the sake of clarity, it should be stated that this is kg NOx as NO2.*

• Thank you. We added this information in Table 1 and in the text.

*Figure 2: the caption should state whether the figure shows the uncalibrated (V1.0) or the calibrated (V1.0A) aCCF.*

• We added this information to the figure captions of figures 2-4.

Figure 2: the figure shows a filament of elevated aCCF for water vapour. I guess this corresponds to a filament of larger PV values indicative of a tropopause folding. If so, the residence time of the water vapour in the stratosphere is short and the non-CO2 effect is probably much less than indicated by the aCCF. What is the confidence level / uncertainty on these values?

Thank you for the interesting question. While it is well established that tropopause folds are the key transport mechanism for stratosphere-troposphere exchange (Holten et al. 1995; Grewe and Dameris 1996), the amount of reversible and irreversible mass exchange with a fold is not. E.g. Holten et al. (1995) states "It is usually found that part of the stratospheric air in the fold returns reversibly to the stratosphere, and part is drawn irreversibly into the troposphere by advection round an anticyclone" and we are not aware that this finding is substantially revised by newer literature. For our application of aCCFs, this implies that a part of the emitted water vapour remains in the stratosphere and is not mixed into the troposphere. This justifies a larger aCCF value compared to the surroundings - outside the fold. Still, the value might be overestimated and may explain a part of the variability in the PV-CCF correlation on which the aCCFs formula is based on (van Manen and Grewe, 2019).

Lines 267-268: why not show or at least give the fraction of aCCF that is zero?

• As the fraction of those areas covered by contrails is varying for different time and date, an overall pdf for contrails would be influenced quite different on individual days, which would make it difficult to compare "non-zero" values of the contrail-aCCF (as this acts as a kind of varying normalisation in the figure). A clear indication of the variation of those "zero-contrail" regions can be seen in Figure 2 (column 3).

Figures 3 and 4: the caption should make it clear that the PDFs are for calibrated aCCF.

• done

Figures 3 and 4: I could not find the % of non-zero aCCF on panels c.

• In Figure 3.c and 4.c we have listed the respective percentage of grid-boxes which are showing zero values (upper left part of the figure c). These values vary between 25-35%.

Figure 5: V1.1 should read V1.0A on top of panel b.

• done

*Line 317: how is AirClim "well established" ?*

• We agree with the reviewer that a more detailed description of the verification of AirClim is missing in the text. We added a paragraph at the end of Sect. 2.3 that aims at proving that AirClim is well established.

Lines 331-334: does this mean that the aCCFs depend on the choice of trajectories for a given set of city pairs? If the calibration depends on the assumption of great circles between city pairs then do the calibration coefficients hold for climate-optimized trajectories?

Thanks for the comment, we think that our explanation was a bit misleading and this requires • further explanation. ACCFs do not depend on the choice of trajectories. First of all, aCCFs have been generated for specific geographic regions: those Lagrangian trajectories, which were used as base simulations for V1.0, were all released in a specific geographic region, namely the North Atlantic Flight corridor. The underlying hypothesis is that due to atmospheric conditions cause such a dependency of aCCFs, derived for specific geographic regions, e.g. mid-latitudes versus tropical latitudes. Hence our alignment study applies aCCFs only in those regions, for which they have been derived (ECAC traffic sample in Europe). Applying aCCFs V1.0 in other regions, is a kind of off-design usage, and we recommend not to use them; alternatively, we highlight the fact that this introduces large uncertainties. We have further clarified our explanation provided here, by first clarifying this "geographic limits where to apply them". In addition, we have also applied aCCFs on three different flight levels in the designated European region, in order to demonstrate our approach. Results show that alignment factors only show a slight variation, which illustrates robustness of alignment factors. We added this information in section 5.3. Third, the influence of the routing strategy on quantitative estimates of non-CO2 effects, requires further explanation. In order to design our alignment study, we had to make sure, that we generate quantitative estimates with the two distinct tools (AirClim, aCCF) which are comparable. Specifically, we had to avoid, that by applying a specific routing strategy, e.g. contrail avoidance, or climate optimized trajectories, we produce an emission inventory of air traffic which on intentionally shows lower contrail effects or lower non-CO2 effects. Such modifications in the trajectories would not enable a consistent comparison of CO2 and individual non-CO2 effects, which is needed when deriving the alignment factors.

**Line 357: can the authors elaborate on the issue here?**

• Thanks for the question. In order to very comprehensively elaborate on any known inconsistencies between the both model setups, we have noted here the different base years for the background meteorology. However, we consider the influence of this effect, to be a higher order effect, with only 10 years difference in the model climatologies (in the wider sense of both "present day" conditions). For comparing to other aviation studies often a linearization approach is applied, as e.g. in Lee et al. when bringing effects from the year 2018 to another base year. It has to be noted, that such scaling approaches assume identical conditions in the background meteorology and processes, hence these studies support our interpretation, to see this as a higher order effect.

*Lines 373-375: I disagree here. A parametrized approach is required for the medium- to long-lived non-CO2 and CO2 effects, but not necessarily for the short-lived effects such as contrails and contrail-cirrus.*

- We agree that a distinction could be made in the methodology applied to long-live CO2 effects versus short lived non-CO2 effects, however, to our understanding, any of these methodologies require parameterizations. The main difference is that for short-term effects a methodology may want to combine a parameterisation approach with a Lagrangian transport approach. However, here, it has to be stressed, that parameterized model representation are always required in order to describe (and quantify) the complex atmospheric responses that takes place, which eventually cause climate effects. These effects can only be represented and quantitatively assessed by parameterized methodologies, as e.g. an explicit solution of the underlying basic physical equations is under most circumstances not possible. A Lagrangian approaches are possible, in particular for short-term effects, however it has to be noted, that also here typical simplications are applied, e.g. only one single Lagrangian trajectory, simplified ice-microphysics, sedimentation, radiation parameterisations, climate sensitivity parameters, etc."
- [Line 479-481] A highly pParameterization approaches are is required in order to develop these response functions and surfaces when studying climate effects due to required assumptions and chosen representation of physical and chemical processes in for such atmospheric comprehensive modelling.

**Lines 418-422: this is indeed a limitation and the associated uncertainty should be quantified.**

- As discussed, we find that spatial resolution requires particular attention when implementing aCCFs. To further stress this issue, we have introduced a subsection in the discussion in order to elaborate on the practical implications of the usage of aCCFs.
- [Line 520] Section 5.2 Practical implications of the usage of aCCFs.

**Line 456: I agree with this statement, therefore the purpose of the manuscript should be to decrease uncertainty not rescale uncertain parameters.**

- Thank you for the comment. The purpose of the manuscript is to allow an alignment of climate estimates of individual non-CO2 effects calculated by aCCF towards absolute strength of individual effect and towards relative strength of individual non-CO2 effects compared to other non-CO2 effects. Hence our alignment factors represent an alternative estimate which could be used in a sensitivity study, in order to explore robustness of estimates of climate effects and associated mitigation gains. We have made that point clearer in the manuscript.
- [Line 562-563] Our analysis illustrates—represents an alternative estimate with absolute individual non-CO2 effects aligned to AirClim V2.0, while also reproducing relative strength of individual non-CO2 effects among each other. [..] As a result, from the alignment<del>calibration</del> performed the updated formulation V1.0A is now available enabling an **alternative** estimation of the climate effects of aviation in order to explore uncertainty range of climate effects and robustness of associated mitigation gains by representing the relative strength of the non-CO2 effects among each other, aligned with state-of-the-art literature.

Line 470ff: This paragraph contradicts lines 72-73 that specify aCCF is also available as an open source Python library of CLIMaCCF V1.0. Why not mention this in the "code availability" section? Or is it not the version used for this manuscript?

• It is a good idea to mention availability of the Python library CLIMaCCF V1.0.

• [Line 580-581 Code availability section] An open source Python library of CLIMaCCF V1.0 is available, which contains an implementation of aCCFs is described in Dietmüller et al. (2023).

Lines 511-514: please provide details of the final version (rather than the submitted version).

• Thanks for pointing this out. The reference has been updated accordingly.

Lines 694-696: please provide details of the final version (rather than the submitted version).

• Thanks for pointing this out. The reference has been updated accordingly.

References: please make sure the format of the references is consistent through the list.

• Thanks for pointing this out. We have revised the format of references accordingly.

Lines 703ff: I find many assumptions to be unnecessarily simplistic, especially those related to thresholds (eg time of day for contrails) or maximum values (eg radiation). Space or time integrals should be more appropriate.

- We understand considerations aiming for a more complex formulation for such algorithmic • climate change functions, which means specifically more dependencies on a larger number of variables, e.g. on how to consider contrails which exist both during day and night. For such day/night-combined contrails, it is obviously even more difficult to provide a reliable estimate of their overall climate effect, which can be composed of warming or cooling effects. The reason for such large uncertainties is that the overall effect also critically depends on their estimated lifetime and their transport pathways, e.g. which duration they exist during day and/or night. This results in large uncertainties in the estimates of their interaction with long-wave radiation as well as short-wave radiation, while noting that the available radiation depends critically also on other atmospheric parameters, e.g. on clouds. In that sense some assumptions are kept on purpose simplistic, e.g. time of day for contrails, and distance to terminator. We estimate that there would be a high risk to create an false confidence, in the case when introducing highly sophisticated thresholds. Because based on our current understanding we do not estimate that current numerical weather forecast would be able to describe a required input parameters with sufficient accuracy. Hence, we judge that it is more thrustworthy to accept limitations, in particular if they are due to large atmospheric variability. Hence, it might be required that one simply needs to accept, that such estimates possess such an extremely large uncertainty, equivalent to large confidence intervals for such estimates (even covering positive and negative signs). However, as we know that such limitations apply in general for the currently existing methodologies in response models, we have included this information in the discussion.
- [Line 484-489] We would like to stress a specific limitation when estimating radiative effects with the help of any parametric response function which relies on numerical weather forecasts. Current methodologies rely on OLR as the basic variable when estimating net contrail effects, however it is known that existence of clouds in the vertical column have a strongly influence a potential climate effect. However, these effects are in general neglected when providing quantitative estimates of the contrail climate effect response functions.

**Line 736: it is not specified how persistent contrails are defined.**

**Line 737: Change Yin et al (2022) to (2023).**

• We have updated the reference Yin et al. (2022) to Yin et al. (2023) in the manuscript.

*Lines 753-754: this is an extreme example of the lack of transparency I mentioned above.*

Fig B1: is that for uncalibrated or calibrated aCCFs?

• Figure B1 was shown for calibrated values. However, there is no Fig B1 anymore in the revised document.

---

## Author Comment (AC3)

*We thank the reviewer for constructive and helpful input and provide our response in detail below.*

**Response to Reviewer Comments**

We thank the reviewers for their careful comments, which improved the quality of the manuscript. Below, the reviewer's comments are repeated in the italic text. Our response follows in normal letters. Blue text is used if text from the revised manuscript is cited. When page and line numbers are specified, they refer to the clean version of the revised manuscript. Additionally, we highlighted the changes in the manuscript by track-change mode, and attached them to the reply.

**REFEREE 1 (RC1)**

*In this study, the authors calibrate a previous version of the algorithmic climate change functions (aCCFs), obtained by fitting climate model simulations to a small selection of meteorological parameters, to the impulse response model AirClim. Both models aim at estimating the average temperature response to CO2 and non-CO2 emissions of flights, but follow different approaches and assumptions. The calibration produces an alternative set of aCCFs, with sizeable differences in the estimated average temperature response of flights compared to the original aCCF set.*

*The study is well written, and the figures and tables illustrate the discussion well. My comments below aim at making the description clearer and more accurate in places and should amount to minor revisions because new analyses are not required.*

**Main comment:**

*Section 3 and 5 need to be more upfront on two important points. First, it is important to point out in section 3 that the "calibration factors" (Table 2) that translate aCCFv1 into functions comparable to AirClim imply very large changes. Those are not small corrections. So the two models, aCCFv1 and AirClim represent two very different views of the climate impact of flights, for the reasons listed (only qualitatively unfortunately) in the conclusion. Second, Section 5 does not answer the basic question of why one would want to calibrate aCCFv1 to AirClim. As a measure of uncertainty? Not really because the two model are unlikely to cover the whole uncertainty range. As a way to choose between different philosophies (tagging/ perturbation and climatological/synoptic)? But what would be the rationale for such a choice?*

- We have reconsidered using the term calibration, and we suggest to replace the term "calibration" with "alignment" in the whole manuscript.
- We are scaling absolute estimates of individual non-$CO_2$ effects (P-ATR20) estimated from aCCFs in a way that they become identical to absolute estimates from AirClim. First, the alignment factors show, which of the two model results in larger P-ATR20 values of the individual effects, exploring absolute estimate. Second, relative strength of individual non-$CO_2$ effects to the overall climate effect gets aligned to the relative strength of AirClim climate estimates, while maintaining with this alignment of aCCF, identical spatial and temporal 'relative distribution' of each individual non-$CO_2$-aCCFs-effect.
- [Lines 19-25] Individual **alignment** factors (i.e. AirClim **alignment**-factors) are provided for the respective non-$CO_2$ effects comprising contrail cirrus, water vapour and $NO_x$-induced climate effects on ozone and methane. **Aligned aCCF-V1.0A can be seen as one realization within the range of possible values.**

Other comments:

*Line 12: "climate effects" Be specific: the aCCFs give the Average Temperature Response.*

- We added the physical climate metric used, by way of example (see track changes in the abstract), as aCCFs can be used to estimate climate effects providing various climate physical climate metrics, e.g. GWP, GTP, ATR.

*Lines 19-20: Clarify: "lower estimates" of the ATR?*

- Thank you for that comment. We have updated the text accordingly, explaining that the aligned aCCFs are aligned in strength and relative strength of individual non-$CO_2$ effects of AirClim V2.0 results (see track changes of revised document).
- **[Lines 21-23] The aCCFV1.0A provide**  **a scaled version of**  aCCF-V1.**0A for climate effects measured as ATR** of all **non-$CO_2$** species **which are aligned for a European air traffic sample to the absolute and relative strength as estimated by AirClim V2.0**.

*Line 49: Not sure that "short-cut" is the right word, because there is also a loss of accuracy. "Approximation" would be a better word.*

- Yes, you're right: Approximation describes this concept well We have updated the text accordingly (see track changes)
- [Line 52-53] Such response surfaces can be seen as **a direct** link and approximation  between aviation emissions and associated climate effects.

*Lines 55 and 62: Need to define what is meant by "perturbation approach" and "tagging contribution approach". It might be useful to add a paragraph before this one to clearly define those two approaches to attribution, and what they mean in terms of radiative forcing. As you found in Grewe et al. (2019) https://doi.org/10.1088/1748-9326/ab5dd7, the impact of the choice of methods is very large: a factor 6.7 according to Table 1 in that paper.*

- We agree that it is useful to the reader to briefly introduce and discuss both approaches. However, we do not agree with the estimated factor of 6.7 when referring to Grewe et al. (2019), By clearly distinguishing the separate updates, table 1 indicates a factor of 1.6 for the climate effect of the ozone signal, and a factor of 2,1 for the combined net NOx-effect. We have added a section in the discussion section exploring the different approaches We included the following text to the revised document:
  [Line 486-473] Overall,  when comparing **the resulting estimates of climate effects from** AirClim **alignment**  factors (Table 2), which align  non-$CO_2$ effect estimates from EMAC/AirTraf/ACCF to the values quantified by AirClim, **we find t**hat **result lie** within the confidence intervals for effective radiative forcing of individual climate effects (when comparing relative variation) which agree with those suggested by assessment studies, e.g., Lee et al., 2021.
  [Line 474-477] Results show that no constant factor is identified, but variations in the order of 20% can be identified. Hence, we consider that changing from perturbation to tagging approach introduces an additional uncertainty in the order of 20%, but does not affect the overall validity of the aCCF concept. With the proposed additional sensitivity study (repeating the workflow for other flight levels), we derive separate set of alignment factors, while exploring vertical dependence of such differences.

*Line 61: I do not understand "for eight specific days". Do you mean eight specific weather situations?*

- Initial baseline simulations that were applied for development of aCCFs covered eight archetypical weather situations; with each being represented by one typical day, hence overall eight individual days. We have further clarified that in the text:
  [Line 67] …for the North Atlantic Flight Corridor (NAFC) for eight specific days **and their synoptic weather situations** that consider representative weather types in summer and winter (Frömming et al., 2021 […].
  [Lines 70-72] The overall concept of aCCFs development relies on statistical methods which link and correlate non-$CO_2$ climate effects as quantified by the **set of** CCFs. Base simulations were performed on explicitly calculated in comprehensive numerical chemistry climate model simulations) [..]

*Line 68: "to local meteorological conditions" is too broad. It is in fact "to a selected set of variables that represent the local meteorological conditions".*

- It is in fact "to a selected set of variables that represent the local physical state of the atmosphere and meteorological conditions".
  **[Lines 72-75] to a selected set of variables that represent the physical state of the atmosphere and** local meteorological conditions at location of emission. Therefore, the strength of **the prototype** aCCF is that their implementation solely requires the local state of the atmosphere as input data**, however this statistical method also introduces uncertainties in the estimated values and represents a simplification.**

*Line 77: Need to clarify what is meant by "effectiveness" here.*

- The term effectiveness here refers to the application of NOx-aCCFs in order to guide flights in such a way, that they avoid regions where high NOx-aCCFs gives large estimates, in order to generate an emission inventory which has an overall lower NOx-induced climate effect. We have added more explanation here for clarity.
- [Line 85-89] For an initial proof of concept **to estimate achievable mitigation gains**, Yin et al. (2018) and Rao et al. (2022) applied an atmospheric chemistry model chain within the EMAC framework to evaluate the effectiveness of NOx-O3 aCCFs using their climate effect information for trajectories optimizations, **hence evaluating if flight guidance which avoids regions with large climate effects as described by aCCF, result in lower overall NOx-induced climate effects.**

*Line 83: The meaning of "the room for a calibration process is open" is only clarified in the next paragraph (line 88) by "one realization within the range of plausible values", so I would suggest merging the two paragraphs to clarify what you are doing. And it should be noted that such calibration does not replace the need for a way to properly account for uncertainty in both AirClim and aCCFs.*

- We agree that we need to describe more in detail, what our study is doing. Hence, we have updated this paragraph accordingly. Additionally, we now suggest to use the term alignment for the factors that we have identified when comparing aCCFs estimates with AirClim estimates.
  The aCCF-V1.**0A** can be seen as **another**  realization within the range of plausible values (event horizon) considering the current level of scientific understanding of climate effects of aviation emissions and their associated uncertainty. The alignment process was performed  by evaluating and comparing the climate effects of aviation for a European  air traffic inventory , resulting from aCCF-V1.0 **(Yin et al. 2023),** with an alternative numerical model; here we used the **advanced** state-of-the-art climate-response model AirClim. A detailed description, visualization, and application of aCCF-V1.0 are provided.

*Line 109: "it is equivalent to selecting another value from this interval". Is that true? If there's a probability distribution, then some values are more probable than others, although I agree that the most probable value is not necessarily the mean.*

- All values within a confidence interval are possible, however, it is correct, that a probability distribution provides additional information if some values are more probable than others, and the mean value is not necessarily the most probable one. We rephrased the text to be more comprehensive.

[Line 123-129] The mathematical formulation given in aCCF-V1.0 (Yin et al. 202**3**) provides a numerical value which represents  individual CO2 and non-CO2 climate effects. Consequently, for our **alignment**  process, we explore , and align**ment factors which would bring** the quantitative estimates **from**  aCCFs to **estimates from an**  alternative modelling approach, i.e.  the climate-response model AirClim.

*Lines 126-127, lines 150-151, line 154: Again, avoid using the imprecise "climate effect". Here, I assume that it is the ATR that is calculated.*

- We intentionally use the term climate effect as we see an equivalence between different physical climate metrics, which can be transferred from one into the other. We agree, that it is helpful to specify, that our study provide estimates in order to scale values that relies on average temperature response (ATR).

*Line 140: "as boundary conditions" – ambiguous. Do you mean that there is no nudging inside the model domain? By the way, is the model configured to simulate Europe only, or is it global?*

- Yes, it is misleading to call our nudging field boundary conditions, hence we have deleted this explanation. We have 4-dimensional fields of atmospheric state variables, and we relax numerically EMAC model conditions towards these values. Thus, we deleted "boundary conditions" (see track changes).

*Line 149: The aCCFs are made out of the CCFs for 8 weather patterns. If I understand well that synoptic information is lost in the aCCFs, so it is not possible to relate aCCFs to specific patterns in the North Atlantic. Is that correct? It would be good to clarify that here. And by the way, are the aCCFs fitted over the 8 weather patterns together? Or are they fitted over some mean of the 8 weather patterns?*

- Yes, aCCFs do not relate to one individual synoptic pattern. Moreover, during their construction of the base simulations, particular attention was paid to cover the set of possible weather patterns by identifying eight archetypical - different - synoptic weather patterns, in order to span the range of possible conditions in the North Atlantic Flight Corridor. For the parameterizations, data from all eight weather-pattern were combined in order to identify a fitting algorithm, which can be applied independently from the prevailing weather situation. We mention the set of CCFs which are used to generate one formulation of aCCFs. This has been clarified in the introduction and further clarified in this methods sections.
  [Line 158-160] Specifically, they allow identifying regions of the atmosphere where aviation emissions induce a strong climate effect, e.g. via the formation of warming contrails or the production of radiatively active species like ozone **independently from the prevailing weather situation.**

*Line 173: Why A330 flights only?*

- For the construction of the emission inventory, we assume that the whole European sample of city pairs is flown by A330 aircraft, as within the EMAC submodel AirTraf corresponding aircraft performance data is integrated for this aircraft.

*Lines 182-186: Again, no need to keep the suspense on the actual calculation by using the imprecise "climate effect". Say that you are calculating P-ATR20 from the start of the paragraph.*

- We agree, that it is helpful to mention the used physical climate metric, P-ATR20. We added some explanation in the revised manuscript now.
  [Line 208-212] While ATR20 or ATR combined with any other time horizon is not addressed in any political framework, it shows advantages for representing aviation related non-CO2 effects (Megill et al. 2024). We note that the short time horizon chosen here, puts more weight on the short-term effects. However, the chosen metric may be customized by constant, species dependent factors to other metrics and other time horizon for both pulse or continuous future emission scenarios similar as in Yin et al. (2023, Tab. 1).

*Caption of Table 1: The "without forcing efficacy" comes as a surprise. There should be an explanation of what that means in the main text.*

- We agree, and we give a very short explanation, but refer to relevant literature in the text now.
- [Line 234] (provided in P-ATR20 in [K] and **including the forcing dependent efficacy values (e.g.; Ponater et al. 2006; Lee et al. 2021)**)

*Line 227: Converting to F-ATR20 when Section 3 has been all about P-ATR20 seems an unnecessarily confusing step. Why do it? And here again, the reader would benefit from a reminder of the notion of efficacy, and how it might change the results compared to Section 3.*

- Thanks for pointing that out. In the revised document we now apply aCCFs with P-ATR20 including efficacy for a day in winter (all figures are changed accordingly), with that we are in line with above alignment chain and moreover contrail aCCFs - which were developed only for typical winter weather patterns (see supplement of Yin et al. 2023) – are applied in the designed area and season.

*Line 245: I would expect variability in the contrail aCCF, but I would also expect some correlation between time steps, since ISSRs are not advected randomly. But then the aCCFs do not have the concept of ISSR, so what do the patterns shown in Figure 2c tell us? It is also surprising to have pockets of cooling in the middle of warming zones – that does not look like the day/night contrast alluded to in the text.*

- Firstly, please note that Fig. 2 in the revised document now displays aCCF patterns for a specific winter condition, as contrails aCCFs were developed for winter weather patterns only.
- aCCFs do integrate the concept of ISSR: we calculate ISSR region every time step using a for ERA5 derived threshold for the relative humidity over ice and a temperature criterion. If ambient air is supersaturated we apply aCCF formulas. We added one sentence in the description of the contrail aCCFs, see Appendix A.
  [Line 920] Note moreover, that contrail aCCFs are only calculated at location and time where persistent contrails are forming, and accordingly regions without persistent contrails are set to zero.

- The intension of Fig. 2c is to shows how ISSR (and associated climate estimate) are varying. Ice supersaturation is a temporarily highly variable field that induces changes within 6h patterns (here pressure level 250 hPa). In some cases, there are ISSR patterns that are quite stationary (see Fig. 2c, 2018-12-02, 0, 6, 12 UTC), here we assume that uplifting of airmasses and connected adiabatic cooling in a stationary system (e.g. connected to a front) induces these ISSR.

- No, this is not day/night contrast, the day/night contrast only can be seen for sunrise and sunset. The current prototype contrail aCCF formulation is a function of the outgoing longwave radiation (see Annex A). Thus, according to the equation, the RF for the daytime contrails can take positive and negative values, depending on the OLR. As the OLR mainly depends on surface temperature and cloud cover, aCCF formulas produce a sharp change in the contrail climate effect estimate in regions where conditions change from clear-sky to cloudy-sky. This explains these pockets of cooling in the middle of warming ones, although we are not sure if these estimates (with strong gradients) are realistic. As mentioned aCCFs are prototypes and we like to stress the limitation mainly of the the day-contrail formulation. In order to address this issue, we added text in the discussion section of the revised manuscript.

- [Line 481] E.g., Yin et al. (2023) discussed several open issues to be addressed further, e.g., the temporal and spatial applicability of the current prototype aCCFs and the uncertainties related to contrail aCCFs. We would like to stress a specific limitation when estimating radiative effects with the help of any parametric response function which relies on numerical weather forecasts. The current algorithm relies on OLR as the basic variable for the estimation of net contrail effects, however no exact information on the existence of clouds in the vertical column is available, e.g. at what altitudes, which is expected to have a non-neglectable influence on the potential climate effect of contrails. However, these effects are in general neglected when providing quantitative estimates of the contrail climate effect response functions.

*Line 307: What is the implication of that narrower distribution for climate-optimised routing?*

- The narrower distribution of the merged non-$CO_2$ aCCF indicates that $NO_x$ effects and contrail effects are getting closer, thus the tradeoff of contrail and $NO_x$ effects will change. Moreover, also the trade-off with $CO_2$ will change significantly (not shown here), as non-$CO_2$ effects are scaled down for aCCF-V1.0A (see Table 2).
  [Line 345] Additionally, the distribution of the merged non-$CO_2$ aCCF is narrower for aCCF-V1.0A, indicating that $NO_x$ effects and contrail effects are getting closer and thus leading to a slightly different trade-off between contrail and $NO_x$ effects. Moreover, also the trade-off with $CO_2$ will change significantly (not shown here), as all non-$CO_2$ effects (unlike for $CO_2$) are scaled down for aCCF-V1.0A (see Table 2).

*Lines 366-367: What do those confidence intervals look like? How do you go from your calibration factors to confidence intervals?*

- Thanks for pointing that out. We know explain more about it in the discussion section. See subsection 5.3: "Uncertainties in prototype aCCFs and in alignment factors" of the revised document.

**Technical comments:**

*Line 26: Suggest rewriting to "long-term background ozone is reduced by the NOx-induced methane decrease".*

- Done

*Line 43: Presenting the three types of studies as bullet point would make that paragraph easier to read.*

- Even though, this is a valid point, we do not like to build points in the introduction, thus we did not follow your advice here.

*Line 88: I have never seen "event horizon" used in this context. Is that a correct use of the term?*

- This term has been used in scenario analysis, in order to describe the "limits" of the possible results and realization. This might not be a familiar expression; hence, we suggest to delete it. [Line 99] *Term "event horizon" has been deleted.*

*Line 115: Typo aligned*

- Done

*Line 117: "is corresponding" -> corresponds*

- Done

*Figure 1: Typo relevant (and it would be good to disable the spelling checker to avoid those words underlined in red)*

- Done

*Line 164: green function -> Green's function*

- Done